# Does training method matter? Evidence for the negative impact of aversive-based methods on companion dog welfare

Ana Catarina Vieira de Castro [1,2,3]*, Danielle Fuchs[1,2,4], Gabriela Munhoz Morello[1,2], Stefania Pastur[1,2,5], Liliana de Sousa[3], I. Anna S. Olsson[1,2]

1 Instituto de Biologia Molecular e Celular, Universidade do Porto, Porto, Portugal, 2 i3S –Instituto de Investigação e Inovação em Saúde, Universidade do Porto, Porto, Portugal, 3 Instituto de Ciências Biomédicas Abel Salazar, Universidade do Porto, Porto, Portugal, 4 Royal (Dick) School of Veterinary Studies, The University of Edinburgh, Edinburgh, United Kingdom, 5 University of Trieste, Trieste, Italy

* ana.castro@ibmc.up.pt

**Data Availability Statement:** All relevant data are within the manuscript and its Supporting Information files.

## Abstract

Dog training methods range broadly from those using mostly positive punishment and negative reinforcement (aversive-based) to those using primarily positive reinforcement (reward-based). Although aversive-based training has been strongly criticized for negatively affecting dog welfare, there is no comprehensive research focusing on companion dogs and mainstream techniques, and most studies rely on owner-reported assessment of training methods and dog behavior. The aim of the present study was to evaluate the effects of aversive- and reward-based training methods on companion dog welfare within and outside the training context. Ninety-two companion dogs were recruited from three reward-based schools (Group Reward, n = 42), and from four aversive-based schools, two using low proportions of aversive-based methods (Group Mixed, n = 22) and two using high proportions of aversive-based methods (Group Aversive, n = 28). For evaluating welfare during training, dogs were video recorded for three sessions and six saliva samples were collected, three at home (baseline levels) and three after training (post-training levels). Video recordings were used to examine the frequency of stress-related behaviors (e.g., lip lick, yawn) and the overall behavioral state of the dog (e.g., tense, relaxed), and saliva samples were analyzed for cortisol concentration. For evaluating welfare outside the training context, dogs participated in a cognitive bias task. Results showed that dogs from Group Aversive displayed more stress-related behaviors, were more frequently in tense and low behavioral states and panted more during training, and exhibited higher post-training increases in cortisol levels than dogs from Group Reward. Additionally, dogs from Group Aversive were more 'pessimistic' in the cognitive bias task than dogs from Group Reward. Dogs from Group Mixed displayed more stress-related behaviors, were more frequently in tense states and panted more during training than dogs from Group Reward. Finally, although Groups Mixed and Aversive did not differ in their performance in the cognitive bias task nor in cortisol levels, the former displayed more stress-related behaviors and was more frequently in tense and low behavioral states. These findings indicate that aversive-based training methods,

**Funding:** The current research study was supported by FCT - Fundação Portuguesa para a Ciência e Tecnologia (Fellowship SFRH/BPD/111509/2015) and UFAW – Universities Federation for Animal Welfare (Grant 14-16/17), with grants awarded to ACVC. SP was supported by PIPOL - Regione Friuli Venezia Giulia. The funders had no role in study design, data collection and analysis, decision to publish or preparation of the manuscript. FCT - Fundação Portuguesa para a Ciência e Tecnologia: https://www.fct.pt/index.phtml.pt UFAW – Universities Federation for Animal Welfare: https://www.ufaw.org.uk/.

**Competing interests:** The authors have declared that no competing interests exist.

especially if used in high proportions, compromise the welfare of companion dogs both within and outside the training context.

## 1. Introduction

To fulfil their increasingly important role as companion animals, dogs need to be trained to behave in a manner appropriate for human households. This includes, for example, learning to eliminate outdoors or walk calmly on a leash [1,2]. Dog behavioral problems are the most frequently cited reason for rehoming or relinquishment of dogs to shelters and for euthanasia [2], which suggests that such training is often missing or unsuccessful.

Dog training most often involves the use of operant conditioning principles, and dog training methods can be classified according to the principles they implement: aversive-based methods use mainly positive punishment and negative reinforcement and reward-based methods rely on positive reinforcement and negative punishment [3]. Within a given training method, several factors may influence how dogs react, such as the characteristics of the behavior under training and the timing of reinforcement/punishment [4]. However, the use of aversive-based training methods *per se* is surrounded by a heated debate, as studies have linked them to compromised dog welfare [5–10]. Some aversive-based tools, such as shock collars, have indeed been legally banned in some countries [11]. However, a recent literature review by [3] concluded that, because of important limitations, existing studies on the topic do not provide adequate data for drawing firm conclusions. Specifically, the authors reported that a considerable proportion of the studies relied upon surveys rather than on objective measures of both training methods and welfare; that they focused on sub-populations of police and laboratory dogs which only represent a small portion of dogs undergoing training; and, finally, that the empirical studies have concentrated mainly on the effects of shock-collar training, which is only one of several tools used in aversive-based training. In summary, limited scientific evidence exists on the effects of the entire range of dog training techniques on companion dog welfare.

Furthermore, previous empirical studies have focused on the effects of training methods on dog welfare within the training context. Behavioral and physiological indicators of welfare, such as the frequency of stress-related behaviors and the concentration of salivary cortisol, have been collected in and around the training situation (e.g., [9,12]; see also [3]). However, the welfare impact of training methods beyond the training scenario has not yet been examined. To our knowledge, only one study evaluated the effects of training on welfare outside the training context. Christiansen et al (2001) [13] found no effect of shock collar training on dog fear or anxiety, but this was based on dog owner reports of behavior and temperament tests rather than on objective and animal-based welfare indicators. Importantly, a suitable assessment of the effects of training methods on dog welfare should comprise an evaluation of their effects both during and beyond the training scenario.

The affective states of animals are influenced by both immediate rewarding or punishing experiences (effects on shorter-term states), and by the cumulative experience of rewarding or punishing experiences (effects on longer-term states) [14]. Hence, due to the repeated exposure to aversive stimuli, training with aversive-based methods is expected to also affect dogs' affective states in a longer-term, transitioning to outside the training context. One way to assess affective states is through the cognitive bias paradigm (e.g., [15]). The cognitive bias task has been validated as an effective tool to evaluate the affective states of non-human animals and has been extensively used with several species, including dogs [16–18]. The rationale

behind the paradigm is based on theoretical and empirical findings that an individual's underlying affective state biases its decision-making and, specifically, that individuals experiencing negative affective states make more 'pessimistic' judgements about ambiguous stimuli than individuals experiencing more positive affective states [14,15,17].

Therefore, the aim of the present study was to perform a comprehensive evaluation of the effects of different training methods on the welfare of companion dogs both within and outside the training context. By performing an objective assessment of training methods (through the direct observation of training sessions) and by using objective measures of welfare (behavioral and physiological data to assess effects during training, and a cognitive bias task to assess effects outside training), we assessed the effects of reward-based and aversive-based methods on companion dog welfare. We hypothesized that dogs trained using aversive-based methods would display higher levels of stress during training, as determined by behavioral and physiological indicators of stress during training sessions, and more 'pessimistic' judgments of ambiguous stimuli during a cognitive bias task performed outside the training context, as compared to dogs trained using reward-based methods. We used a quasi-experimental approach in which dog-owner dyads were recruited to participate through the training school at which they were enrolled. As treatment could not be randomized, data on potential confounders was collected to be included in the analysis of treatment effects.

Understanding the effects of training methods on companion dog welfare has important consequences for both dogs and humans. Both determining and applying those training methods that are less stressful for dogs is a key factor to ensure adequate dog welfare and to capitalize on the human benefits derived from interactions with dogs [19,20].

## 2. Materials and methods

### 2.1. Ethical statement

All procedures were approved by ICBAS (Abel Salazar Biomedical Sciences Institute) ORBEA (Animal Welfare Body). All head trainers of dog training schools and owners completed a consent form authorizing the collection and use of the data.

### 2.2. Training schools

**2.2.1. Recruitment.** Dog training schools within the metropolitan area of Porto, Portugal were searched on the internet. Eight schools were selected based on both their geographical proximity and on the listed training methods. The head trainers were invited by telephone to participate in the study. They were informed that the aim was to evaluate dog stress and welfare in the context of training and the methodological approach was thoroughly explained. To avoid bias during recorded training sessions, the trainers were not made aware that study results were going to be further compared among different training methods. Of the eight contacted schools, seven agreed to participate. After study conclusion, a debriefing with the participating training schools was performed in order to communicate the results.

The training schools had different training sites and class structures. Depending on the school, training sites were located either in rural or urban environments, and classes were conducted either indoors or outdoors. Classes were either individual or in group sessions of 15 to 60 minutes, and varied in frequency and time of day depending on the school: the frequency of classes per week among schools ranged from one to three sessions and classes were taught in the mornings, afternoons or evenings. The type of behaviors trained, on the other hand, was fairly standard across training schools and included teaching the dog to sit, lie down, stay, come when called, not to jump on people and to heel or walk on a loose leash; tricks were also taught in some of the participating schools.

**2.2.2. Classification of training methods.** We performed an objective assessment of the training methods used by each school. To this end, we randomly selected six video recordings of training sessions per training school (see section 2.4.1) and analyzed the videos for the frequency of the intended operant conditioning procedures used for training, namely positive punishment, negative reinforcement, positive reinforcement and negative punishment (see Table 1 for the detailed definitions). In order to be coherent with the standard for classification of operant conditioning procedures as reinforcement or punishment (which is based not on the procedure itself but on its effect on behavior) [21], throughout the paper we refer to the procedures as *intended* positive punishment, *intended* negative punishment, etc. The analysis was performed using The Observer XT software, version 10.1 (Noldus Information Technology Inc, Wageningen, The Netherlands) and afterwards the proportion of intended aversive-based techniques [(number of intended positive punishments + number of intended negative reinforcements)/total number of intended operant conditioning procedures)] was calculated for each training session (see S1 Appendix for the results). Although Schools A, C, D and F all used some form of intended positive punishment and/or negative reinforcement and, as such, their training methods can be classified as aversive-based, the fact that two highly different levels of the use of such techniques were observed lead us to divide these schools in two groups. Schools A and D which used, on average, a proportion of 0.76 and 0.84 of intended aversive-based techniques, respectively, were categorized as Group Aversive, and Schools C and F which used, on average, a proportion of 0.22 and 0.37 of intended aversive-based techniques, respectively, were categorized as Group Mixed. Schools B, E and G, which did not use any intended aversive-based techniques, were classified as Group Reward.

## 2.3. Subjects

The head trainer of each training school was asked to indicate at least fourteen dogs fitting our inclusion criteria (described below), and we then approached the owners to ask if they were willing to participate. The information about the study given to the owners was the same that was given to the head trainers of the schools. The inclusion criteria for the dogs were: 1) to have attended the training school for less than two months, in order to mitigate familiarization to training methods, and 2) to be free of certain behavioral problems (e.g., aggression, fearfulness and separation anxiety, as determined by the owner and the first author), in order to prevent any confounding stress.

**Table 1. Definition of the intended operant conditioning procedures used to classify the dog training schools according to their training methods.**

| Procedure | Definition |
| --- | --- |
| Positive punishment | Any (presumably) unpleasant stimulus that was applied to the dog after the exhibition of an undesirable behavior. These included leash jerks (with either choke or pinch collars), shock delivery through e-collars, slapping the dog, yelling at the dog and leaning towards the dog in a threatening way. |
| Negative reinforcement | Any (presumably) unpleasant stimulus that was applied to the dog and that was stopped only after the dog exhibited the desired behavior. These included pulling the collar upward and releasing the pressure only when the dog sat, pulling the collar downward and releasing the pressure only when the dog laid down, and hanging the dog by the choke collar until he or she calmed down. |
| Positive reinforcement | Any (presumably) pleasant stimulus that was applied to the dog after the exhibition of a desirable behavior. These included food treats, playing tug-of-war, verbal praise, and petting the dog. |
| Negative punishment | Any (presumably) pleasant stimulus that was removed after the exhibition of an undesirable behavior. These involved turning the back to the dog as soon as they jumped or started to mouth, and stopping to walk if the dog was pulling on the leash. |

Over the course of the study, which was conducted between October 2016 and March 2019, the owners of 122 companion dogs agreed to participate. However, 30 dog owners dropped out of the training schools before any meaningful data could be collected. Specifically, these subjects dropped out before meeting our requirement that at least two training sessions were video recorded and that the owner completed a written questionnaire. Consequently, our final sample comprised 92 subjects, 28 from Group Aversive (Schools A and D: n = 14), 22 from Group Mixed (School C: n = 8, School F: n = 14), and 42 from Group Reward (School B and G: 15 dogs, School E: 12 dogs).

As for subjects' demographics, the average age was 11.9 (SEM = ±1.0) months, 54% were male and 35% were neutered/spayed. Thirty-four percent were mixed-breed dogs and the remaining 66% belonged to a FCI-recognized breed group: 18% belonged to Group 1: Sheep-dogs and Cattledogs (except Swiss Cattledogs), 13% to Group 2: Pinscher and Schnauzer–Molossoid and Swiss Mountain and Cattledogs, 5% to Group 3: Terriers, 4% to Group 6: Scent hounds and related breeds, 2% to Group 7: Pointing dogs, 20% to Group 8: Retrievers–Flushing Dogs–Water Dogs, and 3% to Group 9: Companion and Toy Dogs.

## 2.4. Data collection

The study was conducted in two phases. The goal of Phase 1 was to evaluate the welfare of dogs within the training context and the goal of Phase 2 was to evaluate the welfare of these same dogs outside the training context.

**2.4.1. Phase 1 –Evaluating welfare within the training context.** In order to evaluate behavioral indicators of welfare during training, each dog was videotaped for the first 15 minutes of three training sessions using a video camera on a tripod (one Sony Handycam HDR-CX405 and two Sony Handycam DCR-HC23). Five experimenters were responsible for data collection. The cameras were positioned to get an optimal view of the specific participant without interfering with training. The day and time of the training sessions were determined by the training schools and by the participants' availability.

To obtain physiological data on stress during training, six saliva samples were collected per dog to allow assay of salivary cortisol [9,22]. Three samples were collected 20 min after each training session (PT–post-training samples) and three were collected at home on days when no training took place, approximately at the same time as PT samples (BL–baseline samples). Owners were asked not to give their dog water in the 20 minutes preceding each sample collection, nor a full meal in the hour preceding each sample collection, respectively. The timing for sample collections, as well as other recommendations regarding saliva collection for cortisol analysis, were drawn from previous relevant research on dogs' cortisol responses to potentially stressful stimuli [9,22–24]. Owners were instructed on how to properly collect samples of their dog's saliva during the first training session, when the first sample (PT1) was collected by the first author of the study. The following samples were always collected by the owners. A synthetic swab (Salivette®) was rubbed in the dogs' mouth for about 2 minutes to collect saliva. For samples collected at the training schools (PT), the swab was placed back into the provided plastic tube and immediately stored on ice. It was then transferred to a -20°C freezer as soon as possible. For samples collected at home (BL), owners were instructed to place the swab back into the plastic tube and immediately store it in their home freezer. Owners were provided with ice-cube plastic makers to transport the BL samples to the training school during the next scheduled training session without them unfreezing, and they were stored at -20°C as soon as possible. Owners were also provided with detailed written instructions for saliva collection and a phone contact in case any owners had questions related to sample collection. For standardization purposes, we ensured that Phase 1 did not last more than three months for each dog.

**2.4.2. Phase 2—Evaluating welfare outside the training context.** After finishing data collection for Phase 1, dogs participated in Phase 2, which consisted of a spatial cognitive bias task. The end of Phase 1 did not correspond to the conclusion of the training programs for the dogs, as this would result in different dogs being exposed to substantially different amounts of training before being assessed for cognitive bias. Instead, for standardization purposes, we ensured that 1) dogs had attended the training school for at least one month prior to Phase 2 and that 2) the cognitive bias task was conducted within one month of completing Phase 1. Due to limited owner availability, 13 subjects either dropped out or did not meet the criteria for Phase 2, resulting in 79 (24 from Group Aversive, 20 from Group Mixed and 35 from Group Reward) of the original 92 dogs participating in Phase 2. The cognitive bias tasks were scheduled according to owners' availability, both on weekdays and Saturdays.

The test was conducted in an indoor room (7.7 x 3 meters) within a research building at the Abel Salazar Biomedical Sciences Institute (ICBAS), University of Porto in Portugal. All dogs were unfamiliar with the room prior to testing. Two experimenters conducted the test while the dog's owner(s) sat in a chair in a corner area of the room (see Fig 1). Dog owners were asked not to look into the dog's eyes or to speak to the dog during the test, unless the experimenters instructed otherwise. The entire test took place over one meeting for each dog. The room was cleaned with water and liquid detergent at the end of each test.

2.4.2.1. Familiarization period. Prior to the start of the cognitive bias task, the dogs were given the opportunity to familiarize with the test room and the researchers. This consisted of a 10-min period during which the dog was allowed to freely explore the room and engage with the researchers and the owner(s).

2.4.2.2. Training phase. The methodology of Phase 2 was based on [18]. During the training phase, dogs were trained to discriminate between a 'positive' (P) location of a food bowl, which always contained a food reward, and a 'negative' (N) location, which never contained a food reward. At the start of each trial, the dog was held by one trained experimenter—hereafter the 'handler', behind a barrier (2 x 2 m, see Fig 1), while a second trained experimenter—hereafter the 'timer', baited (or did not bait, depending on the type of trial) the bowl with a piece of sausage (approximately 1.25 g for smaller dogs and 2.5 g for larger dogs). To ensure that the

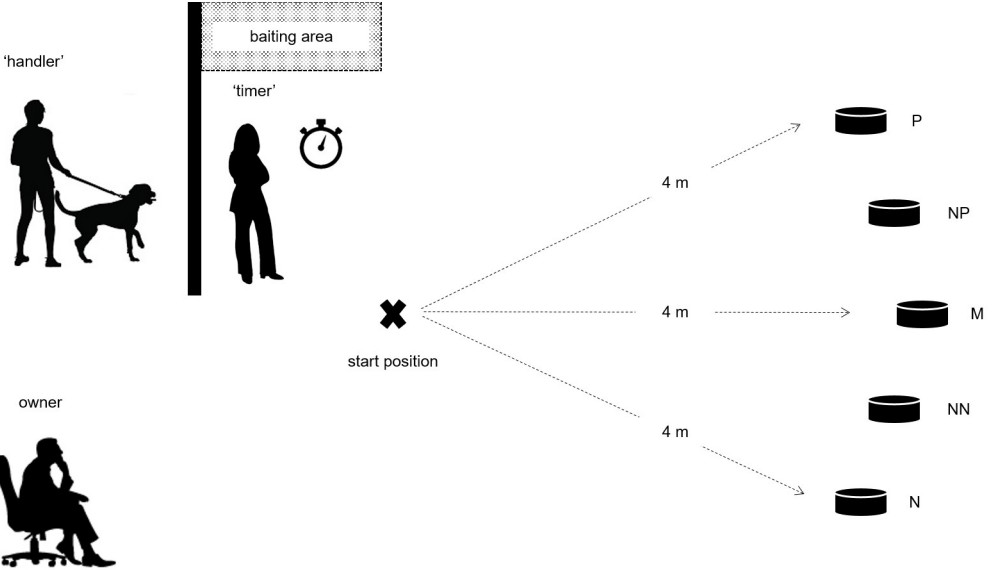

**Fig 1. Schematic representation of the cognitive bias task.**

dog, the owner and the 'handler' were blind to whether or not the bowl contained food during each trial, the bowl was baited out of their sight, on the opposite side of the barrier. Additionally, the food reward was rubbed onto the food bowl before every trial to prevent the influence of olfactory cues. The height of the food bowl was such that the dog could not visually judge the presence or absence of food from the start position.

After baiting (or not baiting) the bowl, the 'timer' placed it at one of the two training locations. The 'timer' then determined the start of the trial, by verbally signaling to the 'handler', upon which the 'handler' led the dog to the start position and released him. The 'handler' always led the dog to the start position on her left side. Because we found that dogs had some difficulty noticing the bowl at the end of the room during pilot tests, the 'handler' walked towards the bowl and pointed it out to the dog during the first four trials. For the remaining trials, the 'handler' simply walked the dog to the start position and released him. After the dog reached the food bowl and (when applicable) ate the reward, the 'handler' collected him and led him behind the barrier to start the next trial. The latency to reach the bowl, defined as the time elapsed between release at the start position and the dog putting its head in line with the edge of the bowl, was recorded for each trial by the 'timer' using a stopwatch.

The position of the 'positive' and 'negative' locations was counterbalanced across subjects and training schools, such that for half of the dogs from each training school, the 'positive' location was on the right hand side as they faced the test area, and for the other half it was on the left. Initially, each dog received two consecutive 'positive' trials (bowl placed in the 'positive' location) followed by two 'negative' trials (bowl placed in the 'negative' location). Subsequently, 'positive' and 'negative' trials were presented in a pseudorandom order, with no more than two trials of the same type being presented consecutively.

All dogs received a minimum of 15 training trials to learn the discrimination between bowl locations. Dogs were considered to have learned an association between bowl location and food (the learning criterion) when, after a minimum of 15 trials, the longest latency to reach the 'positive' location was shorter than any of the latencies to reach the 'negative' location for the preceding three 'positive' trials and the preceding three 'negative' trials. Each trial lasted a maximum of 20 seconds. If the dog did not reach the bowl by that time, the trial automatically ended and a latency of 20 seconds was recorded.

All but two dogs were able to complete the training phase. For the two dogs that failed to complete training, one did not show any interest in the food reward and the other was food-motivated but could not focus on the task. These two dogs belonged to Group Mixed. Therefore, the total number of subjects completing Phase 2 in Group Mixed was 18.

2.4.2.3. Test phase. Testing began once the learning criterion was achieved. Test trials were identical to training trials except that the bowl (empty) was placed at one of three ambiguous locations equally spaced along an arc 4 m from the dog's start position, between the 'positive' and 'negative' locations. The three test locations were: 'near-positive' (NP: one third of the way along the arc from the 'positive' location), 'middle' (M: half way along the arc), 'near-negative' (NN: one third of the way along the arc from the 'negative' location). Three test trials were presented at each test location (nine test trials in total) in the following order for all dogs: M, NP, NN, NP, NN, M, NN, M, NP (each location was presented first, second or third in each block of three test trials). Each test trial was separated from the next one by two training trials identical to those conducted in the training phase (one 'positive' and one 'negative' trials presented in a random order), in order to maintain the associations between the 'positive' and 'negative' locations and the presence or absence of food, respectively. Thus, the test phase included a further sixteen training trials interspersed in blocks of two between the nine test trials.

To end the test phase, a final trial was conducted by placing an empty bowl in the 'positive' location to determine whether dogs ran to the empty bowl as quickly as they did to the baited

bowl. This was meant to establish that the dogs were not relying on olfactory or visual cues during the test. During the entire test, each trial was kept as similar as possible in terms of preparation time and activity, and dogs were handled in the same way throughout the test.

Due to circumstances beyond our control, namely people speaking loudly and other dogs barking in the building during some of the tests, some subjects were clearly distracted and disengaged from the task during some trials. Whenever this happened, no latency was recorded for that trial. The experimenters waited for the dog to resettle and moved to the following trial.

## 2.5. Questionnaire

All owners were asked to complete a brief written questionnaire regarding dog demographics and background, and owner demographics and experience with dogs and dog training. The questionnaire was based on [10].

## 2.6. Data analysis

**2.6.1. Phase 1 –Evaluating welfare within the training context.** 2.6.1.1. Behavior coding. We developed two ethograms based on previous literature to record the frequency of different stress-related behaviors and the time spent in different behavioral states and panting during the training sessions [8,9,23]. The behaviors and their definitions are described in Tables 2 and 3.

Behavior coding was conducted by three observers, which, with the exception of the first author, were blind to the schools' classification based on their training methods. Each video was coded twice, once with the ethogram for stress-related behaviors, using a continuous sampling technique (by the first and second authors, see Table 2), and a second time with the ethogram for overall behavioral state and panting, by scan-sampling at 1 minute intervals (by the first and fourth authors, see Table 3). The Observer XT software, version 10.1 (Noldus Information Technology Inc, Wageningen, The Netherlands) was used to code for stress-related behaviors and Windows Movie Player and Microsoft Excel to code for overall behavioral state and panting.

The second and fourth authors were trained to become familiar with the ethograms and inter-observer reliability was assessed for each ethogram by having the corresponding pair of observers watch and code sets of four videos at an early stage of analysis [9]. Cohen's Kappa coefficient was calculated for each pair of videos using The Observer XT. After analyzing each set of four videos, if there was poor agreement for any video (r<0.80), the observers checked and discussed all the inconsistencies and, if needed, decided on how to refine the description of the ethogram behaviors. After this, they re-analyzed the same videos and the process was repeated until r>0.8 was achieved for the entire set of videos. Once this level was attained, the observers analyzed a new set of four videos. The whole process was repeated until a value of r>0.8 was achieved for the four videos of a new set in the first attempt (i.e., without the need for a re-analysis). At this point, the observers were assumed to be in strong agreement and began coding videos independently [9]. A total of 265 videos were coded. For the ethogram for continuous sampling, the analysis of 16 videos was needed before agreement was achieved, whereas for the ethogram for scan sampling, agreement was achieved only after the analysis of four videos. Afterwards, for each ethogram, the remaining videos were distributed randomly between observers, while ensuring that each observer coded a similar percentage of videos from each experimental group. The first author coded 76% of the videos with the ethogram for stress-related behaviors and 64% with the ethogram for overall behavioral state and panting.

During some training sessions, we were not able to videotape the intended full 15 minutes of training. For these sessions, those that were at least 10-min long were included in the analysis and those that were less than 10-min were excluded.

**Table 2. Ethogram for stress-related behaviors.**

| Behavior | Definition |
|---|---|
| **Avoidance behaviors** | |
| Body turn | Dog rotates its body (or head only) to the side, away from owner/trainer, in an attempt to avoid him/her, following an action such as looking at, approaching, or talking to the dog. Dog is in a tense or low posture. Ears are usually back. Tail can be down. Can be accompanied by lip licking or paw lift. |
| Move away | Dog takes one or a few steps away from owner/trainer (can be with rear or hind paws only), in an attempt to avoid or escape, following an action such as looking at, approaching, or talking to the dog. Dog is in a tense or low posture. Ears are usually back. Tail can be down. Can be accompanied by lip licking or paw lift. |
| Crouch | Dog lowers body (or head only) towards floor, usually lowering its head relative to torso (can be accompanied by blinking and dog's head can generally be turned away), bending legs and arching its back, following an action of owner/trainer, such as looking at, approaching, or talking to the dog. Ears are usually back. Tail can be down. Can be accompanied by lip licking or paw lift. |
| Lying on side or back | Dog lies down on its side or back with the head close to (or in) the ground, in an attempt to avoid confrontation/manipulation by owner/trainer. Legs may be open, exhibiting the ventral region. Movement towards the position is usually slow and gradual. Tail is still and can be curled between the legs. |
| **Vocalizations** | |
| Yelp | Short duration, load, high pitched vocalization. |
| Whine | Long duration, high pitch vocalization. |
| **Other stress-related behaviors** | |
| Fear-related elimination | Expelling of faeces or urine immediately after an action of owner/trainer towards the dog. |
| Salivating | Emitting saliva from the mouth. |
| Body shake | Vigorous movement of whole body side to side. |
| Yawn | Mouth opened wide briefly, then closed (may not close completely). |
| Scratch | Dog scratches itself with mouth or paw. |
| Paw lift | One fore limb only is lifted, usually in a slow movement, and either immediately returned to rest on the ground or remaining lifted for a brief period. It is not directed at any person, animal or object and all other limbs remain on the ground. It is not caused by manipulation from trainer/owner (e.g., pulling the leash upwards) or by the dog adjusting its posture or trying to reach or follow a toy or food. Dog is in a tense or low posture. |

2.6.1.2. Cortisol analysis. Two dogs (one from School B and one from School E, both from Group Reward) did not cooperate with the saliva collection procedure and, as such, no saliva samples were extracted from them. For the remaining 90 dogs, only 23 dog owners (seven from Group Aversive, five from Group Mixed and 11 from Group Reward) were able to appropriately collect six saliva samples. The samples from these subjects were selected for analysis. An additional 40 dog owners (14 from Group Aversive, 11 from Group Mixed and 15 from Group Reward) were able to properly collect at least four saliva samples. From these 40 dogs, eight were randomly selected to have their samples analyzed (one from Group Aversive, three from Group Mixed and four from Group Reward). In total, 8 dogs from Group Aversive, 8 dogs from Group Mixed and 15 dogs from Group Reward had their samples selected for analysis (Schools A, C, D, E and F: n = 4; School B: n = 5; School G: n = 6). These samples were sent to the Faculty of Sport Sciences and Physical Education of the University of Coimbra, Coimbra, Portugal, where they were assayed for cortisol concentration using standard ELISA kits (Salimetrics®).

In order to investigate potential changes in salivary cortisol concentration as a result of training methods, for each dog the baseline sample values (BL) and the post-training sample

**Table 3. Ethogram for overall behavioral state and panting.**

| Overall behavioral state | |
|---|---|
| Tense | Dog shows a combination of: |
| | • horizontal and tense body |
| | • tense muzzle |
| | • ears forward or back |
| | • tail held stiffly (high, neutral or low) and still (in some cases it can be wagging) |
| | • lip licking |
| | • panting |
| | • paw lift |
| | • yawning |
| | • blinking |
| Low | Dog shows a combination of: |
| | • curved body |
| | • bent legs |
| | • low head |
| | • ears back |
| | • tail low, still or wagging |
| | • paw lift |
| | • lip licking |
| | • panting |
| | • yawning |
| Relaxed | Dog shows a combination of: |
| | • horizontal and relaxed body |
| | • ears in the normal position for dog |
| | • tail neutral, still or wagging slightly |
| | • can be panting |
| Excited | Dog shows a combination of: |
| | • horizontal body |
| | • ears forward or back |
| | • tail high or neutral, still or wagging |
| | • rapid or jerky movement |
| | • jumping |
| | • panting |
| | • play signals |
| | • body shake |
| | • may also yawn |
| Unknown | Dog is not visible, video recording is unclear, or the behaviors of the dog cannot be clearly interpreted. |
| **Pant** | |
| Panting | Mouth open, breathing vigorously |
| Not panting | Mouth closed |
| Not visible | Dog's mouth is not visible |

values (PT) were averaged, and the difference between the post-training and the baseline averages computed (hereafter the post-training increase in cortisol concentration).

**2.6.2. Phase 2 –Evaluating welfare outside the training context.** For each dog, we calculated the average latency to reach the food bowl during each of the three types of test trials

(NP, M, NN) as well as the average latency to reach the 'positive' and 'negative' training locations during the test phase.

Seventy-three dogs completed the cognitive bias task. From these 73 subjects, 14 disengaged from the task during some trials due to noise outside the test room. Thirteen disengaged for one test trial (Group Aversive: five dogs at location NP, and three dogs at location NN; Group Reward: one dog at location M, three dogs at location NP, and one dog at location NN), and one (Group Reward) for three test trials (one at each test location). For these dogs, the average latencies to the test locations were calculated from the remaining test trials. Of the remaining four dogs, one (from School G) completed the first seven test trials (at locations M, NP, NN, NP, NN, M, NN), two (one from School A and one from School E) completed the first five test trials (at locations M, NP, NN, NP, NN), and one (from School G) completed the first three test trials (at locations M, NP, NN); then they stopped cooperating with the task. Their average latencies to the test locations were calculated from these trials.

## 2.7. Statistical analyses

Statistical tests were conducted using SPSS® Statistics 25.0 and SAS University Edition®. All data were tested for normality by performing the Shapiro-Wilk test prior to data analysis. Except for the number of trials required to reach the learning criterion in the cognitive bias task, the data were not normally distributed. Thus, Procedure GENMOD was used on SAS University Edition® to perform negative binomial regressions with repeated measures (dog) of all the stress-related behaviors, behavioral states, and panting scans as a function of group (Aversive, Mixed, Reward), training session (Session 1, 2, and 3) and the interaction between these two categorical factors. Procedure GLIMMIX was used on SAS University Edition® to analyze the latency to reach the bowls in the cognitive bias task as a function of bowl location, group, and the interaction between these two categorical variables, considering dog as a random effect. To correct for the unbalanced distribution of potential confounders in the dataset, all known confounders for which sufficient data were available were considered in the analysis as follows. First, each confounder was tested, one at a time, in addition to the variables of interest, to verify if there was a significant relation between confounder and response variable, and if the confounder substantially changed the model estimate of the independent variables (i.e. group and training session estimates). This way of testing confounders, in which they are tested one at a time, allows to maintain enough statistical power to verify their significance and influence in the models. If more than one confounder was found to be significant, then all the significant confounders were tested in the whole model. Non-significant confounders, variables of interest and interactions were removed from the final models. A numeric (dog age) and two categorical factors (children in the household and owner gender) were tested as possible confounders for each stress-related behavior, each behavioral state and panting, as well as for the latencies to reach the bowl in the cognitive bias task. A numeric (dog weight) factor was also tested as a possible confounder in the cognitive bias test. Multicollinearity was checked among confounders and variables of interest prior to testing of statistical models. Least-squared means were compared among groups and training sessions with Bonferroni corrections for multiple pairwise comparisons. Non-parametric tests were used to compare cortisol concentrations among groups, as well as to perform a preliminary assessment of the data, as follows:

- Fisher's exact tests were used to compare the three groups (Aversive, Mixed, Reward) regarding dog demographics and background, and owner demographics and experience with dogs and dog training (variables collected with the questionnaire).

- Kruskal-Wallis tests were used to evaluate the effects of group (Aversive, Mixed and Reward) on the post-training increase in cortisol concentration, on the baseline and the post-training levels of cortisol, as well as on the number of training classes attended by the dogs before Phase 2.

- A Wilcoxon signed-rank test was used to compare, in the cognitive bias task, the latency to reach the P location during test trials and during the final trial, when the bowl contained no food, to verify whether dogs were relying on olfactory or visual cues to discriminate between bowl locations.

Finally, a one-way ANOVA for independent samples was used to compare the number of trials needed to reach the learning criterion in the cognitive bias task among groups (Aversive, Mixed, Reward). All the statistical tests were two-tailed and the level of significance was set at $\alpha = 0.05$. When multiple comparisons were performed, a Bonferroni correction was applied. Specifically, corrected p-values were used for the post-hoc pairwise comparisons performed for the post-training increase in cortisol concentration and for the number of trials to criterion in the cognitive bias task. The effect sizes for all the reported results were calculated as Cohen's d. The entire dataset is available in S2 Appendix.

## 3. Results

### 3.1. Questionnaire

**3.1.1 Dog demographics and background.** Concerning dog demographics, the three groups did not differ in sex and neuter status ratios, but they differed with regards to age ($F = 13.9$, $p = 0.013$) and FCI breed group ($F = 25.3$, $p = 0.008$). As for dog background, the groups differed only in the age of separation from the mother ($F = 20.8$, $p = 0.001$, see Table 4).

**3.1.2. Owner demographics, experience with dogs and dog training.** Regarding owner demographics, the three groups did not differ in owner age and family household size; however, they differed in owner gender ($F = 8.4$, $p = 0.013$) and in whether the household included children ($F = 6.2$, $p = 0.044$). Regarding owner experience with dogs and dog training, the groups did not differ in whether owners had attended training classes with a previous dog nor in whether they had had other dog(s) in the past, but they differed in the information owners used to choose the dog training school ($F = 19.9$, $p = 0.005$, see Table 4).

### 3.2. Phase 1 –Evaluating welfare within the training context

**3.2.1. Behavioral data.** 3.2.1.1. Stress-related behaviors. Dogs from Group Aversive performed significantly more body turn (Group Aversive: (M±SEM) 3.14±0.64 vs. Group Reward: 0.39±0.08; $Z = 7.4$, $p<0.001$, $d = 1.77$), crouch (Group Aversive: 3.56±0.71 vs. Group Reward: 0.59±0.15; $Z = 4.8$, $p<0.001$, $d = 1.10$), body shake (Group Aversive: 1.29±0.19 vs. Group Reward: 0.63±0.10; $Z = 2.8$, $p = 0.014$, $d = 0.66$), yawn (Group Aversive: 2.30±0.28 vs. Group Reward: 0.28±0.07; $Z = 6.6$, $p<0.001$, $d = 1.37$) and lip lick (Group Aversive: 55.90±4.36 vs. Group Reward: 4.11±0.37; $Z = 16.6$, $p<0.001$, $d = 3.91$) than those from Group Reward. Dogs from Group Aversive also performed more yawn (Group Aversive: 2.30±0.28 vs. Group Mixed: 0.80±0.20; $Z = 3.4$, $p = 0.002$, $d = 1.03$), lip lick (Group Aversive: 55.90±4.36 vs. Group Mixed: 17.84±2.15; $Z = 5.7$, $p<0.001$, $d = 1.67$) and tended to perform more body shake (Group Aversive: 1.29±0.19 vs. Group Mixed: 0.64±0.14; $Z = 2.2$, $p = 0.090$) than dogs from Group Mixed. Dogs from Group Mixed performed more body turn (Group Mixed: 2.13±0.39 vs. Group Reward: 0.39±0.08; $Z = 6.4$, $p<0.001$, $d = 1.46$), crouch (Group Mixed: 1.83±0.44 vs. Group Reward: 0.59±0.15; $Z = 3.1$, $p = 0.006$, $d = 0.70$), yawn (Group Mixed: 0.80±0.20 vs. Group Reward: 0.28±0.07; $Z = 2.4$, $p = 0.050$, $d = 0.57$) and lip lick (Group Mixed: 17.84±2.15

**Table 4. Variables obtained from the questionnaire (dog demographics and background, and owner demographics and experience with dogs and dog training).** Fisher's exact tests were used to compare the three Groups (Aversive, Mixed, Reward).

| Variable | Statistical results | Group | | |
|---|---|---|---|---|
| | | Aversive (n) | Mixed (n) | Reward (n) |
| **Dog demographics** | | | | |
| **Breed (FCI)** | F = 25.3, p = 0.008* | | | |
| Mixed breed | | 4 | 7 | 20 |
| Sheepdogs and Cattledogs | | 7 | 6 | 4 |
| Pinscher and Schnauzer | | 7 | 4 | 1 |
| Terriers | | 1 | 2 | 2 |
| Dachshunds | | - | - | - |
| Spitz and primitive types | | - | - | - |
| Scent hounds and related breeds | | 1 | - | 3 |
| Pointing dogs | | 2 | - | - |
| Retrievers, Flushing Dogs and Water Dogs | | 5 | 2 | 11 |
| Companion and Toy Dogs | | 1 | 1 | 1 |
| **Age** | F = 13.9, p = 0.013* | | | |
| younger than 6 months | | 1 | - | 7 |
| 6–11 months | | 10 | 14 | 25 |
| 1–3 years | | 15 | 8 | 9 |
| 4–7 years | | 2 | - | - |
| older than 7 years | | 0 | - | 0 |
| **Sex** | F = 2.7, ns | | | |
| **Neuter status** | F = 0.1, ns | | | |
| **Dog background** | | | | |
| **Dog origin** | F = 16.8, ns | | | |
| **Age of dog when separated from the mother** | F = 20.8, p = 0.001* | | | |
| less than 1 month | | 1 | - | - |
| 1–1.5 months (inclusive) | | 1 | 1 | 9 |
| 1.5–2 months (inclusive) | | 13 | 5 | 11 |
| 2–2.5 months (inclusive) | | 4 | 7 | 6 |
| 2.5–3 months (inclusive) | | 5 | 2 | 2 |
| 3–4 months (inclusive) | | - | 4 | - |
| 4–5 months (inclusive) | | 1 | - | - |
| more than 5 months | | - | - | - |
| don't know | | 3 | 3 | 14 |
| **Other animals at home** | F = 0.2, ns | | | |
| **Age of dog when adopted** | F = 20.7, ns | | | |
| **Age of dog when first taken out** | F = 17.0, ns | | | |
| **Frequency of walks during the first 5 months of life** | F = 22.7, ns | | | |
| **Owner demographics** | | | | |
| **Gender** | F = 8.5, p = 0.013* | | | |
| Male | | 15 | 13 | 11 |
| Female | | 13 | 9 | 31 |
| **Children** | F = 6.2, p = 0.044* | | | |
| Yes | | 15 | 5 | 12 |
| No | | 13 | 17 | 30 |
| **Age** | F = 5.3, ns | | | |
| **Family household size** | F = 5.2, ns | | | |

(*Continued*)

**Table 4.** (Continued)

| Variable | Statistical results | Group | | |
|---|---|---|---|---|
| | | Aversive (n) | Mixed (n) | Reward (n) |
| **Owner experience with dogs and dog training** | | | | |
| **Searched for information before choosing current school** | F = 19.9, p = 0.005* | | | |
| Yes, by visiting dog training schools | | 3 | 1 | 1 |
| Yes, on the different training methods | | 3 | 5 | 13 |
| No, already knew the present school | | 6 | 3 | 2 |
| No, followed third party recommendation | | 8 | 12 | 24 |
| No, the present school was the closest | | 8 | 1 | 2 |
| Other | | - | - | - |
| **Owned other dog** | F = 4.8, ns | | | |
| **Training classes with other dog** | F = 0.2, ns | | | |

*significant differences at α = 0.05.

vs. Group Reward: 4.11±0.37; Z = 7.3, p<0.001, d = 1.99) compared to Group Reward. There was also a tendency for move away to be affected by group ($X^2_{2,265}$ = 5.3, p = 0.073). Overall, when group affected stress-related behaviors, these were performed less frequently in Group Reward than in both Group Aversive and Group Mixed. No effect of group was found for scratch, yelp, whine and paw lift. None of the stress-related behaviors was affected by training session. The average frequencies of stress-related behaviors by group are depicted in Fig 2.

Presence of children in the household was a significant confounder which increased the frequency of body turn (Z = -2.4, p = 0.018), but decreased the frequency of body shake (Z = 2.4, p = 0.016). Additionally, as dog age increased, the frequency of body turn (Z = -2.6, p = 0.011) and yawn (Z = -2.8, p = 0.006) decreased.

There were not enough occurrences of salivating and lying on side/back to perform negative binomial regressions. Salivation frequency was (M±SEM) 0.29±0.14, 0.03±0.03, and 0.02±0.02 in Groups Aversive, Mixed, and Reward, respectively. In a similar pattern, the frequency of lying on side/back was 0.99±0.09 and 0.02±0.02 in Groups Aversive and Mixed, respectively,

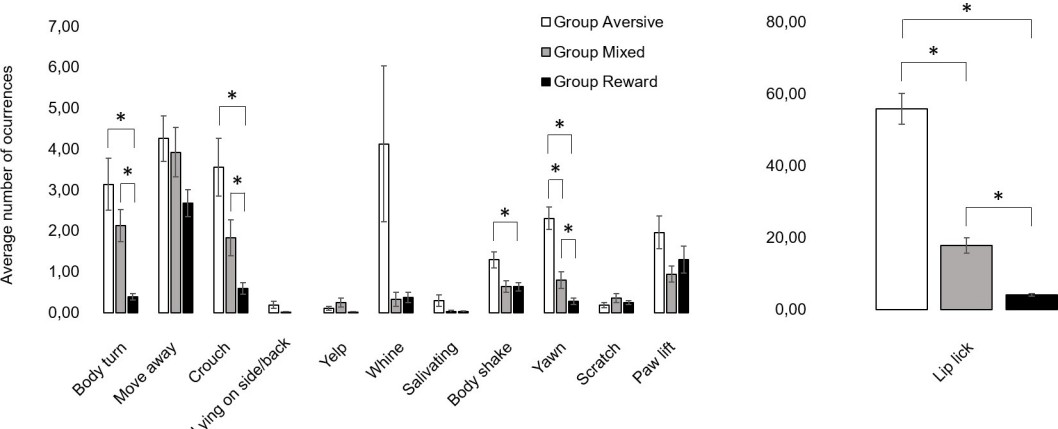

**Fig 2. Number of occurrences of each stress-related behavior averaged across the three training sessions for Group Aversive (white bars), Group Mixed (grey bars) and Group Reward (black bars).** Vertical bars show the SEM. *stands for statistically significant differences for least square means at α = 0.05.

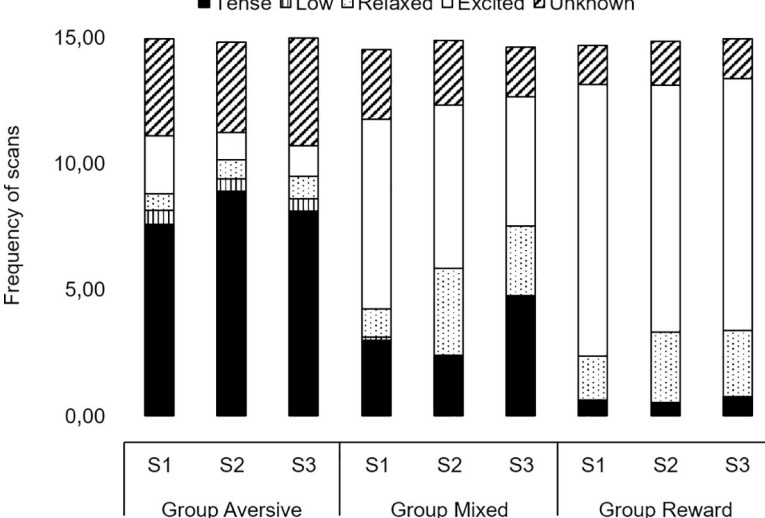

**Fig 3.** Average number of scans in the different behavioral states in training sessions 1 (S1), 2 (S2) and 3 (S3) for Group Aversive (left), Group Mixed (middle) and Group Reward (right).

and no occurrences were observed in Group Reward. Fear-related elimination was never displayed during this study. Model details are available in S3 Appendix.

3.2.1.2. Overall behavioral state. The frequency of scans in which dogs were observed in an excited state was lower in Group Aversive compared to both Group Mixed (Z = 6.2, p<0.001, d = 1.62) and Group Reward (Z = 9.0, p<0.001, d = 2.63). Dogs from Group Mixed were also observed less frequently (Z = -4.2, p<0.001, d = 1.31) in an excited state than those of the Group Reward (Fig 3). Similarly, dogs were observed in a relaxed state less frequently in Group Aversive than in Group Mixed (Z = -2.5, p = 0.033, d = 0.66) and Group Reward (Z = -2.8, p = 0.017, d = 0.78), but no differences were observed between Group Mixed and Group Reward (Z = -0.1, p = 0.999). On the other hand, dogs from Group Aversive were observed in tense and low states more frequently than those from Group Mixed (Z = 5.9, p<0.001, d = 1.85 for tense; Z = 3.7, p<0.001, d = 1.07 for low) and Group Reward (Z = 14.6, p<0.001, d = 2.96 for tense; Z = 3.9, p<0.001, d = 0.81 for low). Dogs were also tense more frequently in Group Mixed compared to Group Reward (Z = 7.6, p<0.001, d = 1.72 see Fig 3). Dogs were observed more often in a low state when children were present in the household (Z = -2.6, p = 0.011).

Training session tended to affect the occurrence of the behavioral state relaxed ($X^2_{2,265}$ = 5.1, p = 0.077) and significantly affected the behavioral state excited ($X^2_{2,265}$ = 10.3, p = 0.006), in that dogs were observed more frequently in an excited state in the first, S1, than in the second, S2, (Z = 2.7, p = 0.019) and in the last, S3, (Z = 3.3, p = 0.003) training sessions.

3.2.1.3. Panting. The frequency of scans in which dogs were observed panting in Group Aversive was higher than in Group Reward (Z = 4.6, p<0.001, d = 1.02). Panting frequency was also observed to be higher in Group Mixed compared to Group Reward (Z = 2.5, p = 0.042, d = 0.59, Fig 4). Training session did not affect the frequency of panting.

**3.2.2. Physiological data.** Baseline cortisol concentrations did not differ among groups [Group Aversive: 0.15±0.02 vs. Group Mixed: 0.14±0.02 vs. Group Reward: 0.13±0.02 µg/dL; H(2) = 1.689, p = 0.430], but differences among groups were found for post-training levels [Group Aversive: 0.26±0.05 vs. Group Mixed: 0.23±0.05 vs. Group Reward: 0.13±0.02 µg/dL; H(2) = 8.634, p = 0.013]. As a result, there was an effect of group in the average post-training increase in cortisol concentration [H(2) = 9.852, p = 0.007]. Specifically, as depicted in Fig 5,

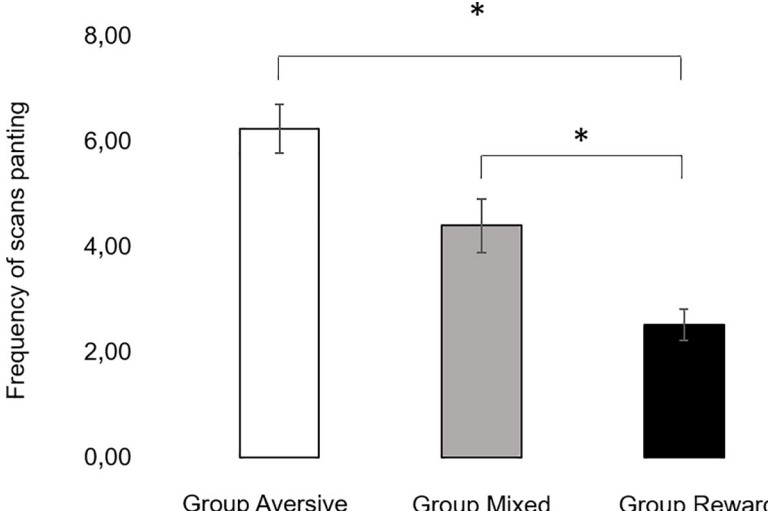

**Fig 4. Number of scans panting averaged across training sessions for Group Aversive (left), Group Mixed (middle) and Group Reward (right).** Vertical bars show the SEM. *stands for statistically significant differences for least square means at α = 0.05.

the average post-training increase in cortisol was higher in Group Aversive than in Group Reward (U = 107.0, p = 0.003, d = 0.13), but no differences were found between Group Mixed and Group Reward (U = 90.0, p = 0.183) nor between Group Mixed and Group Aversive (U = 39.5, p = 0.826).

### 3.3. Phase 2 –Evaluating welfare outside the training context

Prior to the cognitive bias task, dogs from Group Aversive, Mixed, and Reward attended (M ±SEM) 6.29±0.47, 7.14±0.65 and 6.07±0.36 training classes, respectively, with no significant differences observed among groups [H(2) = 2.7, p = 0.258].

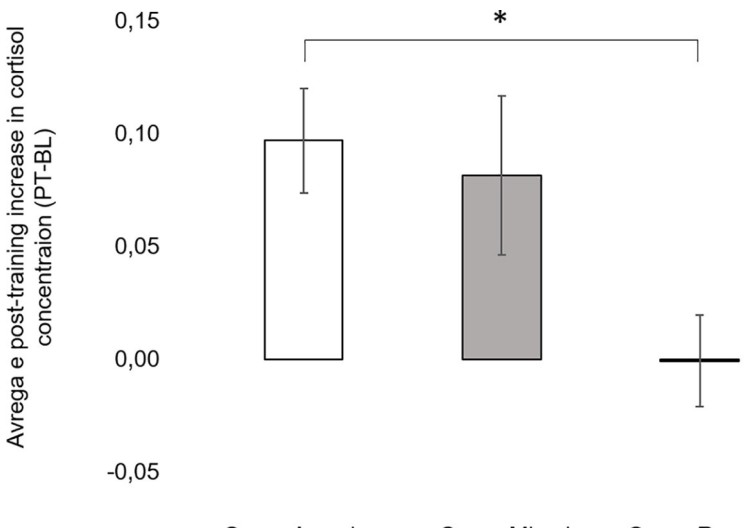

**Fig 5. Average post-training increase in cortisol concentration (PT-BL) for Group Aversive (left), Group Mixed (middle) and Group Reward (right).** Vertical bars show the SEM. *stands for statistically significant differences for the averages at α = 0.05.

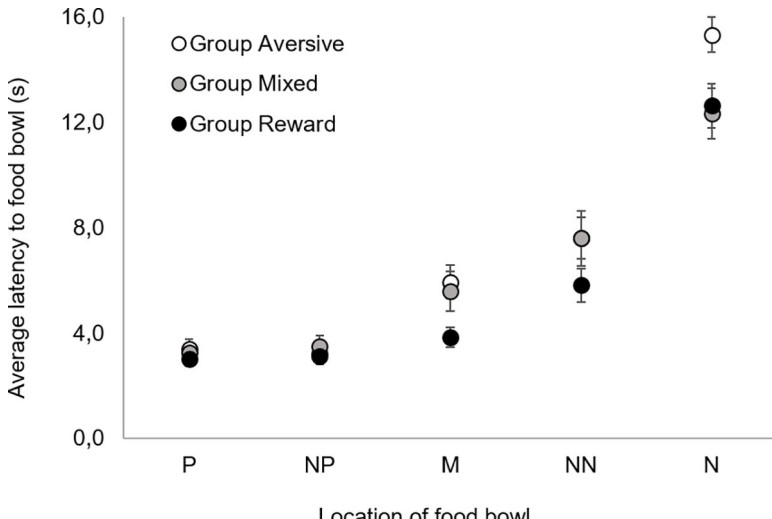

**Fig 6. Average latency to reach the food bowl as a function of bowl location: P—'positive', NP–'near positive', M–'middle', NN–'near negative', N–'negative', for Group Aversive (white circles), Group Mixed (grey circles) and Group Reward (black circles).** Vertical bars show the SEM.

**3.3.1. Training phase.** Dogs took (M±SEM) 27.14±0.85 trials to reach the learning criterion. Dogs from Group Aversive, Group Mixed and Group Reward took, respectively, 28.71 ±1.35, 29.61±1.79 and 24.80±1.26 trials. The differences among groups were statistically significant [F(2,74) = 3.5, p = 0.037], with dogs from Group Mixed showing a tendency to require more trials to reach criterion than Group Reward [t(51) = -2.2, p = 0.090], but no differences being found between Group Aversive and Group Reward [t(57) = -2.1, p = 0.124] nor between Group Aversive and Group Mixed [t(40) = 0.4, p = 0.968].

**3.3.2. Test phase.** Fig 6 depicts the average latencies to reach the two training locations (P, N) and the three test locations (NP, M, NN) for Group Aversive, Group Mixed and Group Reward. Group and bowl location affected the latency for the dogs to reach the bowl, but there was no significant group*bowl location interaction. Dogs of Group Aversive took longer to reach all bowl locations (t = 2.6, p = 0.032, d = 3.99) compared to dogs from Group Reward, but no differences were found between Group Mixed and Group Reward (t = 2.0, p = 0.153), as well as between Group Aversive and Group Mixed (t = 0.4, p = 0.999).

Lastly, an analysis comparing the average latency to reach the P location during test trials and the latency to reach this same location during the final trial (when the bowl contained no food) revealed no significant differences (T = 1295.5, p = 0.328), confirming that the dogs were not relying on olfactory or visual cues to discriminate between bowl locations.

## 4. Discussion

This is the first empirical study to systematically investigate the effects of different training methods on the welfare of companion dogs within and outside the training context. We objectively classified training methods, extended the study of aversive-based methods to other techniques and tools besides shock collars, and used objective and validated measures for the assessment of dog welfare within the training context (behavioral and physiological stress responses during training) and outside the training context (cognitive bias task). Since it became evident during data collection that the recruited dog training schools that employed aversive-based methods did so to a substantially different extent, for analysis the participating schools were divided into three groups: Group Aversive, composed by two schools that used

over 75% of intended aversive-based methods, Group Mixed, composed by two schools that used less than 40% of intended aversive-based methods, and Group Reward, composed by three schools that used no intended aversive stimuli. Overall, our results show that Group Aversive and Group Mixed were in poorer welfare during training than Group Reward, and that Group Aversive was also in poorer welfare than Group Reward outside the training context. Additionally, although no differences between Groups Aversive and Mixed were found outside the training context, Group Aversive displayed poorer welfare during training.

During the welfare assessment in the training sessions, dogs from Group Aversive were observed more frequently in low behavioral states than dogs from Group Reward, and dogs from both Group Aversive and Group Mixed were observed more frequently in tense behavioral states and more frequently panting than dogs from Group Reward. Dogs from Group Aversive were also observed more frequently in tense and low behavioral states than dogs from Group Mixed. Tense and low body postures reflect states of distress and fear in dogs (e.g., [25]), while panting has been associated with acute stress in dogs (e.g., [9,26]). Additionally, overall, dogs from Group Aversive displayed stress-related behaviors more frequently than dogs from both Group Mixed and Group Reward, and dogs from Group Mixed displayed stress-related behaviors more frequently than dogs from Group Reward. In previous studies, high levels of lip licking and yawning behaviors have been consistently associated with acute stress in dogs (e.g., [10,27]). Importantly, lip licking has been associated with stressful social situations [27]. This most likely explains the large magnitude of this behavior observed in Group Mixed and the even larger magnitude in Group Aversive, as aversive-based training methods comprise social and physical confrontation with the dog. The display of avoidance behaviors such as body turn, move away, crouch and lying on side/back, specifically in response to training techniques, highlights the aversive nature of the training sessions at the schools employing aversive-based methods. Notably, lying on side/back was only displayed in Groups Aversive and Mixed (and mostly in School A, which employed the highest proportion of intended aversive-based training methods). Finally, no differences were found among groups for scratch, paw lift, whine and yelp. Previous studies on dog training methods have also failed to identify significant differences in scratch and paw lift [9,10], suggesting that these may not be reliable indicators of stress, at least in the context of training. It is possible that scratching and paw lift behaviors are also associated with excitement and arousal rather than just distress. In turn, whining has also been associated with attention seeking and/or food begging behavior in dogs [28], and as such, is most likely also not a reliable indicator of distress. Finally, yelping may be interpreted as a response to pain [27]. However, besides the fact that no differences were found between groups, this behavior occurred very rarely in the present study. Combined with the observed differences in other stress-related behaviors, this seems to suggest that the aversive-based methods used in the present study caused emotional rather than physical distress. Hence, the present study shows a strong association between the use of aversive-based training methods and an increased frequency of stress behaviors during companion dog training. However, our results also show that the proportion of aversive-based methods used also matters, with lower proportions of aversive stimuli resulting in lower frequencies of stress behaviors exhibited by the dogs. These results strengthen and extend the findings of previous studies on companion dogs, which suggested a positive correlation between the use of both shock collars [9] and other aversive techniques [10] with stress behaviors in the context of dog training.

An effect of training session was found for the behavioral state excited, with dogs being more frequently excited in Session 1 than in Sessions 2 and 3. This result is most likely a consequence of dogs' familiarization with the training context. The tendency of the relaxed behavioral state to increase with training session possibly reflects the reduction in excitement.

With regards to physiological measures of stress, the average post-training increase in cortisol concentration (PT-BL) was higher in Group Aversive than in Group Reward, whereas no differences were found between Group Mixed and Group Reward nor between Group Aversive and Group Mixed. Previous studies investigating cortisol levels in dogs in relation to training have yielded contradictory results. Schalke et al (2007) [29] found significant differences in the cortisol levels of three groups of laboratory dogs trained using shock collars with different degrees of shock predictability (the lower the predictability, the higher the cortisol levels). However, studies comparing aversive- and reward-based training methods have found either no significant differences or the opposite pattern: the effect on cortisol from shock collar and lemon-spray bark collars did not differ from the control treatment [9,30], and a negative punishment training method (a quitting signal) resulted in higher levels of cortisol than the use of a pinch collar (aversive-based technique) [31]. Hence, the present study is the first to report a significant increase in cortisol levels in dogs trained with aversive-based methods as compared to dogs trained with reward-based methods.

The average post-training increase in cortisol levels observed in the present study (M = 0.11 µg/dL for Group Aversive and M = 0.08 µg/dL for Group Mixed) was lower than those reported in other studies that found significant increases after dogs were exposed to aversive stimuli (0.20–0.30 µg/dL in [29] and 0.30–0.40 µg/dL in [23]). One possible explanation for this difference in magnitude may be related to the nature of the stimuli used in the different studies. Whereas the reported elevations in cortisol in [29] and [23] appeared after the presentation of non-social stimuli (shocks in [29], and shocks, sound blasts and a falling bag in [23]), the stimuli used during training in the present study were mainly of a social nature (i.e.: leash jerks, physical manipulation or yelling at the dog). Stimuli administered in a social context may be more predictable or better anticipated and, therefore, generate less acute stress responses [23]. In support of this view, [23] did not find elevations in cortisol after the presentation of social stimuli (physical restraint and opening an umbrella).

When considering welfare outside the training context, we found that, in the cognitive bias task, dogs from Group Aversive displayed higher latencies for all the stimuli than dogs from Group Reward, with no differences being found between Groups Aversive and Mixed nor between Groups Reward and Mixed. Although affect is hypothesized to exert a greater influence on decision-making under ambiguity (i.e., for the test stimuli: NN, M, NP) than under certainty (i.e., for training stimuli: N, P), other studies in cognitive bias have also found differences for both test and training stimuli [e.g., 32–35, see 35 for a review]. This type of result, with differences found for (at least one of) the training stimuli has also been interpreted as evidence for differences in the valence of the affective states. The fact that differences can emerge for both training and test stimuli has been proposed to result from the fact that choice in the cognitive bias task depends on two different components of the decision-making process: perceived probability and perceived valuation of rewards (and punishments). An individual may be less likely to make a less 'risky' or more 'pessimistic' response if they consider the reward to be less probable (or punisher more probable) and/or if they consider the reward to be less valuable (or the punisher more aversive) [35,36]. In summary, affective states may influence the responses to both the training and the test stimuli in the cognitive bias task, although different components of the decision-making process may be playing a role. Therefore, the most likely explanation for the present findings is that dogs from Group Aversive considered the food reward less probable (as indicated by the higher latencies to the test stimuli) and also showed a higher valuation of reward loss relative to win (as indicated by the higher latencies to the training stimuli) [36]. Overall, these results indicate that dogs from Group Aversive were in a less positive affective state than dogs from Group Reward. To our knowledge, the only other study in dogs that addressed the welfare effects of training methods outside the training context was

performed by Christiansen et al (2001) [13], who studied the use of shock collars to train hunting dogs not to attack sheep. No general effects of the use of shock collars on dog fear and anxiety were found one year after training took place. However, unlike the test used by Christiansen et al (2001) [13], which was a modified version of a temperament test used by the Norwegian Kennel Club, the cognitive bias approach used in the current study is a widely established and well-validated method for evaluating animal welfare (e.g., [14–18]). Hence, to our knowledge, this is the first study to reliably assess and report the effects of aversive- and reward-based training methods in the affective states of dogs outside the training context.

Dogs from Group Reward showed a tendency to learn the cognitive bias task faster than dogs from Group Mixed. Similar findings were observed previously by Rooney et al. (2011) [37], who found a positive correlation between the reported use of reward-based training methods and a dog's ability to learn a novel task (touching a spoon with its nose). In another study, Marshall-Pescini et al (2008) [38] found that dogs with high-level training experience were more successful at opening a box to obtain food than dogs which had received either none or only basic training. Although Marshall-Pescini et al (2008) [38] reported that all subjects' training included positive reinforcement methods, they did not specify whether positive punishment and/or negative reinforcement were used in combination. Altogether, previous research suggests that training using positive reinforcement may improve the learning ability of dogs. It remains unclear why a difference was not observed between Group Aversive and Group Reward in the present study. Still, it is important to mention that in all previous studies cited above, animals were required to perform a given behavior in order to obtain a positive reinforcer. Thus, it is unclear whether the same effect would stand if the dogs had to learn a task whose goal was, for example, to perform a behavior to escape from an unpleasant situation. It may be the case that dogs trained with positive reinforcement develop a specific 'learning set' [39] for tasks involving positive reinforcement, but that dogs trained with aversive-based methods perform better in tasks involving some sort of aversive stimuli. Further research is needed to clarify the relationship between training methods and learning ability in dogs.

Notably, we found that the higher the proportion of aversive stimuli used in training, the greater the impact on the welfare of dogs (both within and outside the training context). This result is in line with the findings of a previous survey study, which showed that a higher frequency of punishment was correlated with higher anxiety and fear scores [8]. Still, in the present study, welfare differences were found even when comparing Groups Reward and Mixed, which used a lower proportion of intended aversive-based techniques as compared to Group Aversive. Dogs from Group Mixed showed higher frequencies of stress-related behaviors, were found more frequently in tense states and panted more frequently during training than dogs from Group Reward. When comparing Group Mixed and Group Aversive, the latter showed a higher frequency of stress-related behaviors and was more frequently found in tense and low behavioral states during training. This seems to suggest that, although dogs trained in 'low aversive' schools do not show as many indicators of poor welfare as those trained in 'highly aversive' schools, their welfare may still be at stake.

Moreover, our results suggest that the proportion of aversive stimuli used in training plays a greater role on dogs' stress levels than the specific training tools used. As an example, one school from Group Mixed used pinch and e-collars, whereas another school from Group Aversive only used choke collars during training. Although the tools used by the former school may be perceived as more aversive, the frequency of stress behaviors was higher in dogs being trained at the latter school. The type of (intended) positive reinforcers also appears to be relevant. All schools except the aforementioned school from Group Aversive used primarily food treats as rewards, whereas the latter only used petting. Although this was not the school using

the highest proportion of aversive stimuli, it was the school whose dogs showed the highest frequency of stress behaviors (data not shown). Previous research has shown that petting is a less effective reward than food in training [40]. Having a highly valuable reward might thus be important in reducing stress when aversive stimuli are used in training. The goal of the present study was to test the overall effect on dog welfare of aversive- and reward-based methods as they are used in the real world, but it may be interesting for future studies to focus on disentangling the effects of the different types of stimuli used in training (as has been done with e-collars) [e.g., 9,25].

Finally, some limitations of the present study must be considered. Firstly, because this was a quasi-experimental rather than an experimental study, we cannot infer a true causal relationship between training methods and dog welfare. To do so would require a randomized control trial. However, conducting an experimental study where dogs are designedly subjected to aversive-based methods would raise ethical concerns, as previous studies have already suggested an association between the use of aversive-based methods and indicators of stress in dogs [3, but see 9]. Because we did not randomly allocate dogs to the treatments (training methods), we cannot discard the possibility that there are significant differences between dog-owner pairs that led some owners to choose an aversive-based school and others to choose a reward-based school. There were indeed differences among groups in owner gender, in whether or not the household included children and in the information owners relied on for choosing the dog training school. There were also differences among groups in dog age, FCI breed group and age of separation from the mother. The study was not designed to evaluate the effect of these factors and they were therefore treated as potential confounders in the statistical analysis, in order to account for the possibility that they would affect our results. The effects of training method reported in the study are robust to these confounders. We tested for dog age, presence of children in the household and owner gender, factors which have been shown to potentially affect dog stress and welfare [e.g., 41–44]. The presence of children in the family has been found to be negatively associated with the owners' perception of the relationship with their dogs, in what is to our knowledge the only study addressing how this factor affects dog behavior [43]. Most research into the relationship between dog age and stress indicators has been conducted in senior dogs and consistently shows higher baseline cortisol and higher cortisol responses to stressful stimuli in aged dogs [45,46]; however, our study did not include any senior dog. Schöberl et al (2017) [44] found cortisol to decrease with increasing age of the dog in adult dogs, whereas Henessy et al (1998) [42] found that the juveniles and adults had higher plasma cortisol levels than puppies. Two of the potential confounders were not included in the analysis because of insufficient reliable data: breed (34% mixed breeds, mainly unknown) and age of separation from the mother (22% unknown). Breed differences in behavior are well established [43] but the classification of breeds into groups has not been found to systematically correlate with behavioral similarities [e.g., 47], and the large percentage of mixed breed dogs where the actual breeds were unknown further constrains a meaningful analysis of this factor in our sample. Literature shows that both early [e.g., 48] and late [e.g., 49,50] separation from the mother (before and after 8 weeks-old, respectively) can be associated with stress-related behavioral problems in dogs. Whereas we do not know the animals' stress levels before the start of training, cortisol data shows no differences between training groups on non-training days.

Secondly, a volunteer bias cannot be excluded and hence any generalization of the present results must take this in account. Finally, this study focused on welfare and did not compare the efficacy of training methods. Presently, the scientific literature on the efficacy of the different methodologies is scarce and inconsistent [3]. Whereas some studies suggest a higher efficacy of reward methods [5,12,51–53], one points in the opposite direction [31] and three show

no differences between methods [9,54,55]. This limits the extent of evidence-based recommendations. If reward-based methods are, as the current results show, better for dog welfare than aversive-based methods, and also prove to be more or equally effective to aversive-based methods, there is no doubt that owners and dog professionals should use reward-based training practices. If, on the other hand, aversive-based methods prove to be more effective, the recommendation may be to use aversive stimuli as infrequently as possible during training, and use them in combination with reward-based techniques. This applies not only to training in a formal school setting but whenever owners use reinforcement or punishment in their interactions with the dog.

## 5. Conclusions

Overall, our results show that companion dogs trained with aversive-based methods experienced poorer welfare during training sessions than dogs trained with reward-based methods. Additionally, dogs trained with higher proportions of aversive-based methods experienced poorer welfare outside the training context than dogs trained with reward-based methods. Moreover, whereas different proportions of aversive-based methods did not result in differences in dog welfare outside the training context among aversive-based schools, a higher proportion of aversive-based methods resulted in poorer welfare during training. To our knowledge, this is the first comprehensive and systematic study to evaluate and report the effects of dog training methods on companion dog welfare. Critically, our study points to the fact that the welfare of companion dogs trained with aversive-based methods is at risk, especially if these are used in high proportions.

## Supporting information

**S1 Appendix. Proportion (mean ± standard deviation) of intended aversive-based techniques used during the six training sessions analyzed for each training school.** For each training session, the number of intended positive punishments and negative reinforcements was divided by the total number of intended positive punishments, negative reinforcements, positive reinforcements and negative punishments. Schools A and D were categorized as Group Aversive, Schools C and F as Group Mixed and Schools B, E and G as Group Reward. (DOCX)

**S2 Appendix. Raw data underlying all the analyzes performed in the current research paper.**
(XLSX)

**S3 Appendix. Negative binomial and generalized linear mixed model details.** S3a Table. Analysis of Generalized Estimating Equation for the stress-related behaviors analysis. S3b Table. Analysis of Generalized Estimating Equation for the behavioral state analysis. S3c Table. Analysis of Generalized Estimating Equation for the panting analysis. S3d Table. Solutions for fixed effects from the generalized linear mixed model for the cognitive bias analysis. (DOCX)

## Acknowledgments

We are grateful, first and foremost, to all dogs and their owners who participated in this study; without them this research would never have been possible. A very special acknowledgment to the dog training schools and their trainers that opened their doors for our participant recruitment and data collection.

We would also like to thank Joana Guilherme-Fernandes for the support provided in the development of the setup for the cognitive bias task, for helping with data collection and especially for all the invaluable discussions during study planning and data interpretation. A special acknowledgment also for Margarida Lencastre and Flávia Canastra, who also helped in data collection. We also want to thank Igor M Lopes for the critical help provided with statistical analysis and Jennifer Barrett for input given during data collection and analysis.

Finally, we would like to thank Dr. Carolyn Walsh and the anonymous reviewers for the detailed input on our work, which helped to substantially improve its quality.

## Author Contributions

**Conceptualization:** Ana Catarina Vieira de Castro, Liliana de Sousa, I. Anna S. Olsson.

**Data curation:** Ana Catarina Vieira de Castro.

**Formal analysis:** Ana Catarina Vieira de Castro, Danielle Fuchs, Gabriela Munhoz Morello, Stefania Pastur.

**Funding acquisition:** Ana Catarina Vieira de Castro, I. Anna S. Olsson.

**Investigation:** Ana Catarina Vieira de Castro, Danielle Fuchs, Stefania Pastur.

**Methodology:** Ana Catarina Vieira de Castro.

**Project administration:** Ana Catarina Vieira de Castro, I. Anna S. Olsson.

**Resources:** Liliana de Sousa, I. Anna S. Olsson.

**Supervision:** Ana Catarina Vieira de Castro, Liliana de Sousa, I. Anna S. Olsson.

**Validation:** Ana Catarina Vieira de Castro, I. Anna S. Olsson.

**Visualization:** Ana Catarina Vieira de Castro.

**Writing – original draft:** Ana Catarina Vieira de Castro.

**Writing – review & editing:** Ana Catarina Vieira de Castro, Danielle Fuchs, Gabriela Munhoz Morello, I. Anna S. Olsson.

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
