## [Decision Letter · Decision Letter 0]

16 Dec 2019

PONE-D-19-29749

Does training method matter?: Evidence for the negative impact of aversive-based methods on companion dog welfare

PLOS ONE

Dear Dr Vieira de Castro,

Thank you for submitting your manuscript to PLOS ONE. After careful consideration, we feel that it has merit but does not fully meet PLOS ONE’s publication criteria as it currently stands. Therefore, we invite you to submit a revised version of the manuscript that addresses the points raised during the review process.

Thank-you for your research on this important topic regarding dog training methods and individual welfare. I agree with the two reviewers that this topic is valuable; however, I also agree with their suggestions for improving the manuscript. This will involve some significant revisions, but I would encourage you to take on these re-writing requirements with the perspective that they are intended to allow your findings to have the strongest impact possible. 

Before addressing the specific reviewer comments that you need to consider, however, I first need to state my concern about a lack of clarity around your process for recruiting dog training schools to participate in your study, and whether fully informed and on-going consent was possible for the head trainers/dog school management. While you declare in your Ethics Statement that “All head trainers of dog training schools and owners completed a consent form authorizing the collection and use of data”, your statement on lines 110-117 indicates that you used partial disclosure or partial deception when recruiting the dog training schools, since you did not reveal that the aim of the study was to compare the effects of different training methods. I believe that the use of partial disclosure could be justified during the recruitment process to avoid biasing the sample; however, the general principles of ethical conduct for research involving humans is that following the use of partial disclosure or deception, a debriefing be carried out, and the participants allowed the opportunity to withdraw their consent/involvement/data after it has been collected. Although this particular circumstance is different from those that typically involve humans as participants in experiments, I think a case could be made in which the head trainers/business owners claim that their lack of knowledge about the study’s purpose might increase their risk from continued participation in the study. That is, particularly in light of the manner in which you categorize training schools as “aversive” and “reward-based”, and the details provided about these schools in Appendix S1, Tables S1a and S1c, it is not unfeasible that particular training schools could be identified and suffer economic consequences (e.g., lower enrolment in future classes). I have brought this to the attention of the PLoS One editorial staff and the Staff Editors require you to provide further information about the Ethics process you engaged in (please see “Request from Editorial Staff” copied below). 

Apart from the Ethics information we require, here are additional revisions for you to address;

1) Statistical Methods:

Reviewer 1 has raised some concerns around your choice of statistical methods. Although your data are non-normally distributed for the most part, which I understand to be the reason you use non-parametric statistics, the reviewer points out- using the example of “panting”-  that such statistics preclude the ability to evaluate interactions among your variables. I would ask you to consider whether you could use more sophisticated statistics that would allow you to examine possible interactions and, if not, please further justify your choice. As well, as the reviewer points out, the sheer number of statistical tests run greatly inflates your risk of committing Type II errors. While I noted that you mention using Bonferonni corrections in Table S3b, there appears to be no use of a correction for multiple comparisons elsewhere- e.g., when reporting your significant correlations. There are many other ways to address the problem of multiple testing (False Discovery Rate, etc.)- please consider adopting one such method. 

2. Training Categories

Both Reviewers point out that there may be issues with your categorization of the methods/schools as “aversive” vs. “rewards-based”- please see their specific comments below. While your Table S1c shows the total number of “aversive” techniques used, with the schools identified as “reward-based” showing none of these, you do not differentiate between the schools on the basis of relative amount of positive reinforcement used by these “aversive” schools. As well, when such techniques are used (timing during learning) could affect stress in the dogs. While you address some of this in your Discussion, I would like you to consider how you might be able to integrate this more fully into your results and interpretation of the results. 

Related to the above, please provide some more clarity around the 4 videotaped sessions you used to categorize the schools’ training methods. Did this occur prior to the beginning of dog/owner recruitment/video-recording? In your results, when you correlate the number of aversive stimuli used by schools with behaviour, cortisol, etc., are you using the data collected during these sessions? If so, have you considered using data on the number of aversives used during training for your actual subjects? Why or why not? I could argue that you would make a much stronger case if you had measures of the numbers of aversives used per dog in each school to correlate with the dog’s behaviours/cortisol. Please comment on this.

3. Cognitive Bias Test

Reviewer 1 suggests that you have possibly over-interpreted the CB test results and I tend to agree. This test can be influenced by both acute and long-term stressors. As suggested, please add some information to your Discussion around interpretation issues of the test itself. 

Even if you effectively argue for interpreting the CB test results as indicators of long-term welfare, this is unlikely due to welfare based solely on training methods used earlier. This is an over-simplification since you clearly show that your two groups differ in other characteristics that might be expected to influence cognitive bias- e.g., as in Appendix S3, bred, age, age when separated from mother, owner/main trainer gender, and whether more than one dog lives in the home. These factors may impact measures of welfare (especially those you call “long term”, i.e., occurring outside the training context). This should be clearly acknowledged and the result interpretation modified.

As well, in Figure 7, the data point for the Reward Group “N” position appears to be missing.

4. Cortisol measures

 My own experience with cortisol is that it is a “messy” hormone and interpretation can be a challenge. One of this issues with your averaging of cortisol levels by group is that the time of day training classes occurred and saliva samples were taken appears to vary widely among training schools/dogs. Please clarify if you think this had no effect, and why – for example, if the time of day for saliva collection between dogs in each group was about the same, then you could argue it had little effect. I do congratulate you on getting multiple samples for most of your dogs. However, your cortisol analysis is based on what might be considered a relatively small sample size (16 vs. 15 per group), and power is likely an issue as well. Is there any relationship for the individual dogs sampled with the specific number of aversives used in training for those dogs? I suspect that the result reported on lines 543-544 uses the average number of aversives per school? Please comment. 

As well, Figure 5 is not compelling. Instead, the data on baseline vs. post-training CORT levels for the two groups would be better shown in the Figure.

5. Terminology

Reviewer 2 raises an important point about how you use the terms “positive punishment” and “negative reinforcement”. Please consider how best to address this, since its not likely that you have data to show that the techniques were actually reinforcing or punishing behaviour, based on their consequences. This will likely require re-wording throughout the manuscript and in Table S1b. 

I will note here that your Ethograms were fantastic!

6. Inter-rater Reliability

I commend you on the efforts you made to standardize behavioural coding across your three observers. However, it appears that, following training – which gave a high Cohen’s Kappa coefficient- the vast majority of videos were coded by the only observer (ACVC) who was not blind to the dogs’ condition/group. Following training, some percentage (~20% seems to be standard) of the videos coded by ACVC should have been coded by at least one other observer, and inter-rater reliability (e.g., intra-class correlation) for specific behaviours provided. If this was done, please provide these data. If this was not done, there is some concern that implicit bias from the “non-blind” coder could influence the data, and this must be addressed. 

I hope you find these reviews and editorial comments to be helpful to you. I look forward to your responses.

Please see further requirements from the Editorial Staff below (Additional Editor Comments).

We would appreciate receiving your revised manuscript by Jan 27 2020 11:59PM. To enhance the reproducibility of your results, we recommend that if applicable you deposit your laboratory protocols in protocols.io, where a protocol can be assigned its own identifier (DOI) such that it can be cited independently in the future. For instructions see: http://journals.plos.org/plosone/s/submission-guidelines#loc-laboratory-protocols

We look forward to receiving your revised manuscript.

Kind regards,

Carolyn J Walsh, PhD

Academic Editor

PLOS ONE

Additional Editor Comments:

***Request from the Editorial Staff:

For research involving human subjects, PLOS ONE expects prior approval by an institutional review board or equivalent ethics committee, and reporting of details on how informed consent for the research was obtained (https://journals.plos.org/plosone/s/human-subjects-research).

We noticed that you obtained ethics approval for your study and consent from the head trainers of dog training schools and dog owners for the collection and use of the data. We also noticed that you did not reveal the full purpose of the study to the dog training schools during the recruiting (lines 113-117). We are uncertain whether your ethics approval covered this partial disclosure of the purposes of the study.

Please could you clarify if your ethics approval covered your research with human participants, in particular, the partial disclosure of the purposes of the study. If ethical approval was not required, please provide a clear statement of this and the reason why, and any relevant regulations under which the work is exempt from the requirement for approval. If the ethics approval was waived by your ethics committee, please provide a copy of this documentation formally confirming that ethical approval was not needed in this case, in the original language and in English translation as supporting information files. This is for internal use only and will not be published.

We kindly request you please clarify all the above concerns in your revision of the manuscript

Case Number: 06481122

ref:_00DU0Ifis._5004P10cD3s:ref

Journal Requirements:

**When submitting your revision, we need you to address these additional requirements:**

**Please ensure that your manuscript meets PLOS ONE's style requirements, including those for file naming. The PLOS ONE style templates can be found at http://www.plosone.org/attachments/PLOSOne_formatting_sample_main_body.pdf and http://www.plosone.org/attachments/PLOSOne_formatting_sample_title_authors_affiliations.pdf**

Reviewers' comments:

Reviewer's Responses to Questions

**Comments to the Author**

1. Is the manuscript technically sound, and do the data support the conclusions?

Reviewer #1: Partly

Reviewer #2: Yes

2. Has the statistical analysis been performed appropriately and rigorously? 

Reviewer #1: No

Reviewer #2: Yes

3. Have the authors made all data underlying the findings in their manuscript fully available?

Reviewer #1: Yes

Reviewer #2: Yes

4. Is the manuscript presented in an intelligible fashion and written in standard English?

Reviewer #1: Yes

Reviewer #2: Yes

5. Review Comments to the Author

Reviewer #1: Interesting study that advances the knowledge of training methods & is worth publishing. Nice to extend observations outside the training ring. That said, there are several overarching concerns:

The number of pairwise tests without corrections & the resulting inflated risk of Type II errors. Further, nonparametric tests such as pairwise comparisons & Friedman's don't allow for consideration of interactions. This is especially concerning as the authors discuss effects of training session on outcomes specific to groups (i.e. it appears there are indeed interactions, which makes discussion of main effects inappropriate). At a minimum, alpha corrections or multiple comparisons should be included. More elegant statistical approaches would benefit the manuscript. For example, 'panting' could have been considered using logistic regression with training group, training session, & group*session as independent variables.

The (over)interpretation of the judgement bias paradigm used to infer long-term welfare effects. It would be nice to have the same or similar parameters compared both acutely and long-term (i.e. long term effects on cortisol & behavioral indicators), although I appreciate there are limitations with that approach. Cognitive bias paradigms are widely used as the authors state, but they are also widely over-interpreted. If one refers back to the original Mendl paper & the sources cited within, it becomes readily apparent that judgement biases can be influenced by *acute* stressors as well as long-term stress. That means it is not de facto evidence of long term affective states. The discussion does a nice job considering nuance in relation to other elements of the study, but presents no discussion of difficulties with interpretation of cognitive bias paradigms. This needs to be added to the discussion.

Finally, lumping all schools that use *any* aversives together oversimplifies the approaches used by different training schools. While the reward based training schools used *no* aversives, there is a strong argument to be made that mainly positive punishment differs from truly balanced approaches that use as much or more positive reinforcement as aversives when training operant behaviors but also employ aversives when proofing already learned behaviors. The authors do discuss this nicely in lines 673-698, but this should be considered in the interpretation of the findings. The methods, per se, may not cause welfare problems but if used unpredictably (e.g. as found in the Schalke et al (2007) study) or in the absence of instruction during acquisition phases may lead to stress during training.

Reviewer #2: It was a pleasure to review “Does training method matter?: Evidence for the negative impact of aversive-based methods on companion dog welfare.” In this study the authors evaluated short and long term effects on aversive and reward based training methods on companion dog welfare. They tested 92 dogs, 42 were tested in reward base training facilities and 50 were tested from aversive based training facilities.

I really enjoyed the concept of this paper. I think investigating training methods is a very important topic. Little is known about the impacts of training methods overall. And specifically the difference between aversive and reward based methods even though these methods have been around for decades and promoted and televised by many different groups. However, I have some minor revisions for this manuscript.

General Comments:

I would argue that the terms positive reinforcement, negative reinforcement and positive punishment are used incorrectly throughout this paper. In Table S1b the authors define Positive Punishment as an “unpleasant stimulus applied to the dog” and Negative reinforcement as an “unpleasant stimulus that was applied and stopped”…”. However, in the field of behavior analysis, were these terms originated, reinforcement and punishment are not necessarily unpleasant or pleasant stimuli as viewed by an observer. For a stimulus to be considered punishment it must decrease a behavior and for it to be considered reinforcement it must increase a behavior. In this study data wasn’t collected on what happened to the behavior after these methods were implemented, thus we don’t know if punishment or reinforcement was used. I would suggest that the authors change these terms to something other than reinforcement and punishment so that it is not confusing to the reader.

The authors stated in the methods that they collected the frequency of the methods and labeled the school aversive or reward based on the frequency that they observed. Those that used more aversive methods were labeled aversive and those that used more rewards were labeled rewarding. It would be helpful if we can see these frequency and a list of specific methods for the aversive and reward schools so that the reader has a better idea of how aversive and rewarding these schools were determined. I find this very important as this is what the whole paper is based on.

Methods

Line 140 “free of behavioral problems…” should clarify “ free of certain behavioral problems” or you could use specific instead of certain.

Line 116 the authors state that the first 15 minutes of three training sessions were used. Were these training sessions randomly picked? If so how many training sessions did the dog have during this study that you could have chosen from?

6. PLOS authors have the option to publish the peer review history of their article (what does this mean?). If published, this will include your full peer review and any attached files.

Reviewer #1: No

Reviewer #2: No

---

## [Author Response · Author response to Decision Letter 0]

8 Mar 2020

RESPONSE TO REVIEWERS

We appreciate all the constructive criticism provided by the editor and the two anonymous reviewers. We have reconsidered the statistical analysis of our data as outlined in more detail in the answer to the specific comments below. Briefly, this resulted in a significant revision which lead us to include Dr. Gabriela M Morello as co-author. Following your considerations, we also decided to alter data analysis and presentation to include three training groups. Specifically, in the current version of the manuscript, we sub-divided the aversive-based schools in two groups according to the relative proportion of aversive based methodologies used: Group Aversive, which includes the two schools using the highest levels of aversive stimuli and Group Mixed, which includes the two schools using the lowest proportions of aversive stimuli. We believe that this alteration allows for a more detailed interpretation of the actual effects of different training methods on dog welfare and significantly improves our paper. 

In what follows, we present detailed responses to all the comments.

Ethics approval information

First, we wish to clarify the situation regarding ethical approval for the study. The study was designed to measure the behavior of the dogs and not that of the owners or trainers. Accordingly, the request for approval was sent to the institutional animal ethics and welfare body, which would normally raise the question of additional ethics approval if they perceive this to be relevant, but did not do so in this case. The fact that human behavior data is not included in the ethics approval for this study also means that it is not possible for us to collect data on aversive stimuli used for each of the research subjects during each of the training session. Taking into account your relevant point that the study implies partial disclosure as regards the training schools, we are now debriefing the schools. It is of course very important to avoid that any individual school is identifiable in the paper, and for that reason we have removed details in Appendix S1 which may be unique for a school (such as when training takes place) but is not relevant data for the research paper. Our study included seven schools from a region in which there are about twenty dog training schools, and after we have edited out the details in S1 it is not possible to identify any individual school from the data we have provided.

Editor Comments

1. Statistical Methods

1.1. I would ask you to consider whether you could use more sophisticated statistics that would allow you to examine possible interactions

In the current version of the manuscript, all the behavioral data (stress-related behaviors, overall behavioral state and panting) was analyzed with negative binomial regressions, with Group (Aversive, Mixed, Reward), training session (Session 1, Session 2, Session 3) and Group*Training session as factors.

1.2. As well, as the reviewer points out, the sheer number of statistical tests run greatly inflates your risk of committing Type II errors. While I noted that you mention using Bonferonni corrections in Table S3b, there appears to be no use of a correction for multiple comparisons elsewhere- e.g., when reporting your significant correlations.

In order to clarify where Bonferonni corrections were used we added in the Statistical analyses section “When multiple comparisons were performed, α was set at α=0.05/number of comparisons (Bonferroni correction). Specifically, post-hoc pairwise comparisons were performed for the post training variation in cortisol concentration, the number of trials to criterion and the adjusted latencies to the ‘middle’ location in the cognitive bias task with α=0.017. Post-hoc pairwise comparisons were also performed for the effects of dog-related variables in the dependent variables measured in the present study (see Appendix S3)” (lines 420-425).

2. Training categories

2.1. you do not differentiate between the schools on the basis of relative amount of positive reinforcement used by these “aversive” schools.

The current version of the manuscript includes this information. Namely, we analyzed the number of “intended” positive reinforcers and negative punishers in addition to the number of “intended” positive punishers and negative reinforcers, and then calculated the proportion of “intended” aversive-based techniques: (number of intended positive punishments + number of intended negative reinforcements)/total number of intended operant conditioning procedures. We recognized that this analysis was more informative for presenting and interpreting the differences between the different aversive-based schools.

2.2. As well, when such techniques are used (timing during learning) could affect stress in the dogs.

We entirely agree with editor in this comment and this was the reason why we controlled for this factor in our study. In our inclusion criteria, all dogs had “to have attended the training school for less than two months, in order to mitigate habituation to training methods”. With this, we intended to ensure that all dogs were in an initial (and similar) phase of learning.

2.3. While you address some of this in your Discussion, I would like you to consider how you might be able to integrate this more fully into your results and interpretation of the results. 

We agree with both the Editor and Reviewer 1 in that grouping schools that use highly different proportions of aversive-based techniques during training was an oversimplification of the issue under study. Hence, in the current version of the manuscript, the original Group Aversive was further divided in two groups: Group Aversive (composed by schools A and D, which had the two highest proportions of intended aversive-based techniques: 0.76 and 0.84, respectively), and Group Mixed (composed by schools C and F, which had the two intermediate proportions of intended aversive-based techniques: 0.22 and 0.37, respectively). 

2.4. Related to the above, please provide some more clarity around the 4 videotaped sessions you used to categorize the schools’ training methods. Did this occur prior to the beginning of dog/owner recruitment/video-recording? In your results, when you correlate the number of aversive stimuli used by schools with behaviour, cortisol, etc., are you using the data collected during these sessions? If so, have you considered using data on the number of aversives used during training for your actual subjects? Why or why not? I could argue that you would make a much stronger case if you had measures of the numbers of aversives used per dog in each school to correlate with the dog’s behaviours/cortisol. Please comment on this.

We are now aware that our original description was not sufficiently clear. The videotaped sessions used to categorize the training methods were randomly selected among all videotaped sessions and were thus used as actual data for the present study. We hope to have made it sufficiently clear in the current version of the manuscript. In lines 118-121, one can now read “We performed an objective assessment of the training methods used by each school. To this end, we randomly selected six video recordings of training sessions per training school (see below, section 2.4.1) and analyzed the videos for the frequency of the intended operant conditioning procedures utilized during training”. In order to make our case stronger we decided to analyze two more sessions per training school; hence the six instead of four training sessions. However, the addition of two extra training sessions did not alter the degree of aversive stimuli used in each school found in the original version of the paper.

In the present version of the manuscript, as we opted for dividing the dogs in three rather than two groups, we no longer have correlation analyzes between the frequency of aversive stimuli and the different welfare indicators; the new division already takes into account the different degrees of the use of aversive stimuli. Yet, we would like to clarify that in the previous version of the paper we did correlate the different welfare measures with the number of aversive stimuli used by the training schools during the 4 (at the time) randomly picked training sessions. Although we agree that a much stronger case would be made if we had the number of aversive stimuli used per dog in each school, there were ethical reasons behind the decision of not doing so. The study was designed to measure the behavior of the dogs and not that of the owners or trainers. Accordingly, we only obtained ethical approval from the institutional animal ethics and welfare body. The fact that human behavior data is not included in the ethics approval for this study means that it is not possible for us to collect data on aversive stimuli on the level of individual owner-dog pairs.

3. Cognitive Bias Test

3.1. Reviewer 1 suggests that you have possibly over-interpreted the CB test results and I tend to agree. This test can be influenced by both acute and long-term stressors. As suggested, please add some information to your Discussion around interpretation issues of the test itself. Even if you effectively argue for interpreting the CB test results as indicators of long-term welfare, this is unlikely due to welfare based solely on training methods used earlier. This is an over-simplification since you clearly show that your two groups differ in other characteristics that might be expected to influence cognitive bias- e.g., as in Appendix S3, bred, age, age when separated from mother, owner/main trainer gender, and whether more than one dog lives in the home. These factors may impact measures of welfare (especially those you call “long term”, i.e., occurring outside the training context). This should be clearly acknowledged and the result interpretation modified.

We agree with both the editor and Reviewer 1 and changed the text to reflect the fact that the differences we found between groups as regards demographic data can have also influenced the results and that both short- and long-term stressors can influence the cognitive bias results. This was included in the end of the Discussion section, where now it can be read “Because we did not randomly allocate dogs to the treatments (training methods), we cannot discard the possibility that there are significant differences between dog-owner pairs that lead some owners to choose an aversive-based school and others to choose a reward-based school. There were indeed differences among Groups in owner gender, in whether or not the household included children and in the information owners relied on for choosing the dog training school. There were also differences among Groups in dog age, breed and age of separation.” (lines 690-695) and “Thirdly, some caution is advised when interpreting cognitive bias results as unambiguous indicators of long-term welfare as they can be influenced both by short- and long-term stressors [37].” (lines 700-702)

3.2. As well, in Figure 6, the data point for the Reward Group “N” position appears to be missing.

We need to clarify that there was no data point missing in Figure 6. In this figure, the data points for the three groups overlap for the N and P positions due to the adjusted latency score we used, which set P as 0 and N as 1 for all subjects. In the current version of the manuscript, this information was added to the figure caption.

4. Cortisol measures

4.1. One of this issues with your averaging of cortisol levels by group is that the time of day training classes occurred and saliva samples were taken appears to vary widely among training schools/dogs. Please clarify if you think this had no effect, and why – for example, if the time of day for saliva collection between dogs in each group was about the same, then you could argue it had little effect.” 

We agree with the editor in that the time of day (which indeed varied among dogs and training schools) could have affected our cortisol levels. However, we corrected for this potential confounder when we used and analyzed the difference between PT and BL levels for each dog instead of the absolute PT and BL levels. This is also the reason why our Figure 5 reports the average difference between PT and BL levels instead of the absolute levels. Moreover, each individual dog always had saliva collected at the same time of the day for PT as well as BL.

4.2. I do congratulate you on getting multiple samples for most of your dogs. However, your cortisol analysis is based on what might be considered a relatively small sample size (16 vs. 15 per group), and power is likely an issue as well.

Although we agree that our cortisol analysis seems to be based on a relatively small sample size, during study design we performed sample size calculations. For salivary cortisol concentration these revealed that, for a power of 80%, the recommended sample size was n=5 per group, which is lower than our current sample size. 

4.3. Is there any relationship for the individual dogs sampled with the specific number of aversives used in training for those dogs? I suspect that the result reported on lines 543-544 uses the average number of aversives per school? Please comment.

The results did indeed refer to the average number of aversive stimuli used in each school, but please refer to section 2.4 above.

4.4. As well, Figure 5 is not compelling. Instead, the data on baseline vs. post-training CORT levels for the two groups would be better shown in the Figure.

Please refer to section 4.1 above.

5. Terminology

Reviewer 2 raises an important point about how you use the terms “positive punishment” and “negative reinforcement”. Please consider how best to address this, since its not likely that you have data to show that the techniques were actually reinforcing or punishing behaviour, based on their consequences. This will likely require re-wording throughout the manuscript and in Table S1b. 

We agree with the editor and Reviewer 2 and in the current version of the manuscript we changed the terminology to “intended positive punishment”, “intended negative reinforcement”, etc.

6. Inter-rater Reliability

I commend you on the efforts you made to standardize behavioural coding across your three observers. However, it appears that, following training – which gave a high Cohen’s Kappa coefficient- the vast majority of videos were coded by the only observer (ACVC) who was not blind to the dogs’ condition/group. Following training, some percentage (~20% seems to be standard) of the videos coded by ACVC should have been coded by at least one other observer, and inter-rater reliability (e.g., intra-class correlation) for specific behaviours provided. If this was done, please provide these data. If this was not done, there is some concern that implicit bias from the “non-blind” coder could influence the data, and this must be addressed. 

Once again, we acknowledge we were not clear enough in the first version of the manuscript. Training involved coding of data from actual subjects, namely four videos for the ethogram for scan sampling and sixteen videos for the ethogram for continuous sampling. Hence, some percentage (although not 20%) of the actual data was coded by two observers. Namely, 1,5% of the data analyzed with the scan sampled ethogram was coded by ACVC and SP and 6% of the data analyzed with the continuous sampling ethogram was coded by ACVC and DF.

Reviewer #1 Comments

1. The number of pairwise tests without corrections & the resulting inflated risk of Type II errors. Further, nonparametric tests such as pairwise comparisons & Friedman's don't allow for consideration of interactions. This is especially concerning as the authors discuss effects of training session on outcomes specific to groups (i.e. it appears there are indeed interactions, which makes discussion of main effects inappropriate). At a minimum, alpha corrections or multiple comparisons should be included. More elegant statistical approaches would benefit the manuscript. For example, 'panting' could have been considered using logistic regression with training group, training session, & group*session as independent variables.

Please refer above to section 1 of Editor Comments.

2. The (over)interpretation of the judgement bias paradigm used to infer long-term welfare effects. It would be nice to have the same or similar parameters compared both acutely and long-term (i.e. long term effects on cortisol & behavioral indicators), although I appreciate there are limitations with that approach. Cognitive bias paradigms are widely used as the authors state, but they are also widely over-interpreted. If one refers back to the original Mendl paper & the sources cited within, it becomes readily apparent that judgement biases can be influenced by *acute* stressors as well as long-term stress. That means it is not de facto evidence of long term affective states. The discussion does a nice job considering nuance in relation to other elements of the study, but presents no discussion of difficulties with interpretation of cognitive bias paradigms. This needs to be added to the discussion.

Please refer above to section 3.1 of Editor Comments.

3. Finally, lumping all schools that use *any* aversives together oversimplifies the approaches used by different training schools. While the reward based training schools used *no* aversives, there is a strong argument to be made that mainly positive punishment differs from truly balanced approaches that use as much or more positive reinforcement as aversives when training operant behaviors but also employ aversives when proofing already learned behaviors. The authors do discuss this nicely in lines 673-698, but this should be considered in the interpretation of the findings. 

Please refer above to section 2.3 of Editor Comments.

Reviewer #2 Comments

1. I would argue that the terms positive reinforcement, negative reinforcement and positive punishment are used incorrectly throughout this paper. In Table S1b the authors define Positive Punishment as an “unpleasant stimulus applied to the dog” and Negative reinforcement as an “unpleasant stimulus that was applied and stopped”… However, in the field of behavior analysis, were these terms originated, reinforcement and punishment are not necessarily unpleasant or pleasant stimuli as viewed by an observer. For a stimulus to be considered punishment it must decrease a behavior and for it to be considered reinforcement it must increase a behavior. In this study data wasn’t collected on what happened to the behavior after these methods were implemented, thus we don’t know if punishment or reinforcement was used. I would suggest that the authors change these terms to something other than reinforcement and punishment so that it is not confusing to the reader.

Please refer above to section 5 of Editor Comments.

2. The authors stated in the methods that they collected the frequency of the methods and labeled the school aversive or reward based on the frequency that they observed. Those that used more aversive methods were labeled aversive and those that used more rewards were labeled rewarding. It would be helpful if we can see this frequency and a list of specific methods for the aversive and reward schools so that the reader has a better idea of how aversive and rewarding these schools were determined. I find this very important, as this is what the whole paper is based on.

 We believe that Table S1a in the present version of the manuscript contains the information requested by the reviewer. Additionally, in section 2.2.2. Classification of training methods of the paper we describe in detail how the analysis of training methods was carried out.

3. Line 140 “free of behavioral problems…” should clarify “free of certain behavioral problems” or you could use specific instead of certain.

In the current version of the manuscript one can read “free of certain behavioral problems”. (line 166)

4.

4.1. Line 116: the authors state that the first 15 minutes of three training sessions were used. Were these training sessions randomly picked?

 The three training sessions were chosen to occur early in training for each. As part of our inclusion criteria, and as reported in the paper, the first session needed to happen before the dog completed two months of training at the training school and the last session ought to occur within three months of the first one. Apart from that, which sessions were picked was mainly determined by the availability of the owner and the experimenters, which were collecting data at several schools at the same time. 

4.2. If so how many training sessions did the dog have during this study that you could have chosen from?

Because the frequency of training sessions differed between schools (and even between dog owners within the same school), we decided to choose the time in training rather than the number of training sessions to define our criteria. Answering to the reviewer’s question in terms of time of training, the three-videotaped training sessions happened somewhere in-between three weeks and three months of training.

---

## [Decision Letter · Decision Letter 1]

15 Apr 2020

PONE-D-19-29749R1

Does training method matter? Evidence for the negative impact of aversive-based methods on companion dog welfare

PLOS ONE

Dear Dr Vieira de Castro,

Thank you for submitting your manuscript to PLOS ONE. After careful consideration, we feel that it has merit but does not fully meet PLOS ONE’s publication criteria as it currently stands. Therefore, we invite you to submit a revised version of the manuscript that addresses the points raised during the review process.

Dear Ana,

Thank-you for the revisions to your manuscript and the clarification about the ethical procedures for the study. I think that your statement from your reply regarding ‘de-briefing’ the instructors/owners at the training schools regarding the true goal of the study should be included in your section on Recruitment (2.2.1). 

I truly appreciate you and your co-authors intentions for conducting this study, and I wish to see it published. However, despite the very high quality data collection, analyses, and writing, I feel that the study suffers from one fundamental issue with which I am having much difficulty. This is, in part, why I have been delaying my decision letter, as I have been trying to assess what the best course of action might be. 

Specifically, my major issue is with the presentation of the study:  its overall treatment is as if the study is an experiment - even though you very clearly acknowledge in the Discussion that it is not. Despite this acknowledgement of the non-experimental design, however (along with an acknowledgement that other factors may have influenced your outcomes), you have evaluated the group differences in cortisol change and cognitive bias using “training school approach” as if it is the ONLY significant difference between the three groups. And, as you very transparently state, it is not. Because dogs/owners were NOT randomly assigned to the training schools, these multiple pre-existing differences between the groups could explain both your cort and CB findings just as well as the Training School differences can, but have not been assessed in any way.

In my opinion, most relevant among these factors are the significant BETWEEN-Group differences in:

1) Breed composition-  in the Group Reward, 74% of the dogs are mixed breed or Retrievers vs. lower percentages of these ‘breed groups’ in the Aversive and Mixed Groups;

2) Age of the dogs – e.g., in Group Reward, 76% are under 1 year old vs. only 39% in Group Aversive;

3) Owner gender- In Group Reward, 74% of owners are Female vs. 46% and 40% in Groups Aversive and Mixed, respectively; and

4) Presence of children in the house- about three-quarters of owners in each of Groups Reward and Mixed do NOT have children vs. about half who do in Group Aversive (which also might simply be a proxy for differences in owner age and lifestyle?).

Although I appreciate that you spend considerable effort evaluating for the effects of these differences WITHIN groups, due to the small sample sizes for the categories within each training type group, there is little statistical power to detect any such effects. Thus, this is a weak test. As well, there are suggestions in the literature that some of these factors MIGHT be viable explanations for your findings (e.g., lower stress in dogs with women handlers; breed differences in temperament, etc. ).

So, as it currently stands, the fundamental problem that the manuscript has is that you are actually unable to actually dismiss any of the above competing ‘alternative’ explanations to ‘training school approach’ as explanations for the differences in your welfare assessment measures. Indeed, it would be fair to criticize your approach as biased in your view that ONLY training approach differences ‘created’ the welfare differences. Although this is absolutely fine to have as your hypothesis, because of the study design, it cannot be the only differences that is evaluated. Until you can convincingly show that the welfare differences you found are based on training school approach as the most compelling explanation, then your study will fall short of being able to advocate for the position that reward-based training generates the best welfare outcomes for dogs.  

Personally (as a dog owner and agility/rally enthusiast who uses only R+ methods), I very much want your conclusion to be the case!! As well, my understanding of learning theory tells me that your prediction about the relationship between training approach and welfare is likely to hold true. However, in evaluating this work as a researcher and academic editor, I believe that the confounds among your group have to be given equal (statistical) consideration before we can suggest that, in this quasi-experimental design, your conclusion is justified. I agree with you that it will be rare for any canine researcher to be able to run a randomized controlled trial with pet/companion dogs for ethical reasons. So, it is important for us to make the strongest and most unbiased case possible based on study designs such as your current one. 

My decision for your manuscript is “Major Revision” based on the above concerns. I do not consider myself a statistician, so I would suggest that you consult with one about this broader issue of how to effectively address the confound issues. Whatever the statistical technique used, I believe you have to convincingly show that the BEST explanation for differences in your welfare measures is NOT breed, age of dogs at testing, owner gender, or other lifestyle aspects, but IS training school approach to which the dogs have been exposed. If this turns out not to be the case- for example, if some of these other differences do explain the welfare measure differences equally as well as the training school approach, then this is also very important to publish, as it will advance our knowledge. Of course, such findings would involve re-framing the paper significantly. 

I hope you find this critique helpful. I believe your work is valuable and needs to be published in a way that leaves little room for reproach or criticism about the study’s findings.  As well, please review all the comments from Reviewer #1 and Reviewer #3 (new). I believe you have adequately dealt with the part of comment #1 of Reviewer #3 regarding baseline cortisol measures (although, as the reviewer correctly points out, there are no ‘baseline’ measures prior to each training session- and this should be acknowledged in the paper); however, the other suggestions in comment # 1 as well as the other comments (particularly # 3) should be considered further. 

I invite you to revisit your manuscript in light of my comments and those of the Reviewers and respond to the above decision.  

Best,

Carolyn

We would appreciate receiving your revised manuscript by May 30 2020 11:59PM. To enhance the reproducibility of your results, we recommend that if applicable you deposit your laboratory protocols in protocols.io, where a protocol can be assigned its own identifier (DOI) such that it can be cited independently in the future. For instructions see: http://journals.plos.org/plosone/s/submission-guidelines#loc-laboratory-protocols

We look forward to receiving your revised manuscript.

Kind regards,

Carolyn J Walsh, PhD

Academic Editor

PLOS ONE

Reviewers' comments:

Reviewer's Responses to Questions

**Comments to the Author**

1. If the authors have adequately addressed your comments raised in a previous round of review and you feel that this manuscript is now acceptable for publication, you may indicate that here to bypass the “Comments to the Author” section, enter your conflict of interest statement in the “Confidential to Editor” section, and submit your "Accept" recommendation.

Reviewer #1: (No Response)

Reviewer #3: (No Response)

2. Is the manuscript technically sound, and do the data support the conclusions?

Reviewer #1: Partly

Reviewer #3: Partly

3. Has the statistical analysis been performed appropriately and rigorously? 

Reviewer #1: No

Reviewer #3: Yes

4. Have the authors made all data underlying the findings in their manuscript fully available?

Reviewer #1: Yes

Reviewer #3: Yes

5. Is the manuscript presented in an intelligible fashion and written in standard English?

Reviewer #1: Yes

Reviewer #3: Yes

6. Review Comments to the Author

Reviewer #1: The authors did a nice job editing the text for language & overhauling the statistics section. I'm puzzled why a mixed model negative binomial regression was not used to control for the random effects of individual? This is a repeated measures design so partitioning that error is appropriate.

Though they have made many changes to the manuscript, the cognitive bias issue has not been addressed in a substantive way. Because so much is made of assessing long-term welfare, I think it demands a more detailed examination of why they chose to substitute an entirely novel welfare indicator for Phase II without including any of the behavioral or physiological indicators used in Phase I. It bears discussing how a possibly transient negative affective state indicates long-term poor welfare.

This is a new comment, but I suggest including means + SEMs in the text of the results section. The graphs are difficult to see, and it seems odd to have to go the supplemental materials to view results used as dependent variables. A summary table with average occurrence by Group would be helpful. It would be nice if the Discussion considered the uniformly low occurrence of stress indicators per training session... while statistically significant one wonders how biologically relevant some of these differences are (e.g. 3 versus 4 "move away")? Similarly I think the authors have captured nice data worthy of further consideration - e.g. what do they tell us about potentially robust indicators of acute stress?

Reviewer #3: I have now read the manuscript entitled “Does training method matter? Evidence for the negative

impact of aversive-based methods on companion dog welfare” by Vieira de Castro, Fuchs, Pastur, Morello, de Sousa & Olsson. The study investigates the effect of aversive- and reward-based training on short- and long-term welfare of dogs. The authors grouped the dogs in three categories according to the prevalence of aversive-based methods in the training - i.e, the Group Aversive, the Group Mixed and the Group Reward. They studied short-term welfare scoring the stress-related behaviours during the training. They also compared the amount of cortisol in dogs’ saliva at home and after training. Finally, they studied long-term welfare using a Judgement Bias paradigm, which assessed the affective state of the dogs outside the training context.

Results show that stress-related behaviours were prevalent in both the Group Aversive and the Group Mixed. The cortisol in dogs' saliva increased during the training only in the Group Aversive. Both the Group Aversive and the Group Mixed showed a pessimistic-like judgement bias.

The findings are not surprising given the previous literature on this topic, but this study is one of the few combining different objective measures of stress to address the relationship between training methods and dogs’ welfare. Given the importance of companion dogs in our daily life, it is therefore important to study the appropriate conditions to train dogs while preserving their welfare.

Major concerns:

1. Authors found that the stress-related behaviours changed across the three groups during the training. But we do not know what the dogs’ level of stress was before training. It is possible, therefore, that other factors may have led to stress-related behaviours. Consider for instance the living conditions of the dogs. Owners that choose specific training centres may have different approaches to the dogs when interacting with them in their home context. Moreover, several factors in training methods influence dogs’ performance and possibly their level of stress, like a tight schedule of reinforcement/punishment, the characteristics of the to-be-punished behaviour, and several other features. There is plenty of literature on the topic (I just mention one review chapter, but there are several others: Hineline, P. N., & Rosales-Ruiz, J. (2013). Behavior in relation to aversive events: Punishment and negative reinforcement. In G. J. Madden, W. V. Dube, T. D. Hackenberg, G. P. Hanley, & K. A. Lattal (Eds.), APA handbooks in psychology®. APA handbook of behavior analysis, Vol. 1. Methods and principles (p. 483–512)) which seems not only to be missing in the Introduction but also in the Authors’ hypothesis, as revealed by the absence of a baseline measurement for each participating dog.

2. The non-parametric test used for the Judgment Bias Paradigm does not allow to test the Group x Bowl Location interaction. This interaction would attest that the latency to approach the different locations of the bowl changed according to the training conditions. Therefore, it provides stronger evidence in support of your hypothesis. This interaction should be explored. On the topic, see Gygax (2014, doi: 10.1016/j.anbehav.2014.06.013).

3. It is not clear to me whether the dogs had finished the training before doing the Judgment Bias task or they had just completed phase 1 of your experiment. If they had not completed the training, then it is misleading to claim that the Judgment Bias task assessed long-term welfare. In general, I think that discussing the results in terms of welfare within and outside the training context would be more appropriate because “short-“ and “long-term” are ambiguous concepts.

Minor points:

4. Could you please report the effect sizes of your results to facilitate future meta-analysis on this topic?

5. Line 165: “… in order to mitigate familiarization to training methods…”;

6. Line 456: “… and a tendency for body turn…”;

7. Line 495: “… there was a tendency for…”;

8. Line 602: “… methods used also matters,…”;

9. Line 609: “This result is most likely a consequence of dogs’ familiarization with the training context…”;

10. Line 690: “… was a quasi-experimental rather than experimental study…”.

7. PLOS authors have the option to publish the peer review history of their article (what does this mean?). If published, this will include your full peer review and any attached files.

Reviewer #1: No

Reviewer #3: No

---

## [Author Response · Author response to Decision Letter 1]

31 May 2020

RESPONSE TO EDITOR AND REVIEWERS

We appreciate all the constructive criticism provided once again by the editor and the two anonymous reviewers. In our opinion, the manuscript as it currently stands has improved in both data analysis quality and strength of results. Generally, we included and tested confounders in our statistical models, re-analyzed the cognitive bias data with a modeling approach where we were able to test for an interaction between latency and bowl location, and reconsidered our claims on having assessed long-term welfare - in the current version of the manuscript cognitive bias is now treated as an indicator of welfare outside the training context. In what follows, we present detailed responses to all the comments.

Editor Comments

1) Until you can convincingly show that the welfare differences you found are based on training school approach as the most compelling explanation, then your study will fall short of being able to advocate for the position that reward-based training generates the best welfare outcomes for dogs. 

As a consequence of not being able to randomly assign dogs to training schools, as you pointed out, we did end up with a dataset with some unbalanced distribution of possible confounders. We agree that it is important to show that the welfare differences are explained by the training school approach and are not artefacts of this distribution. We have addressed this by taking the confounders into account in the statistical analysis in the current version of the manuscript. We tested each confounder first, one at a time (i.e. 1 additional degree of freedom being used) in addition to our variables of interest, to verify if they were significantly related with the response variable and also if they substantially changed the independent variables’ estimates of the models (i.e. group and training session estimates). This way of testing confounders, in which they are tested one at a time, allows to maintain enough statistical power to verify their significance and influence in the models. If more than one confounder was found to be significant, then all the significant confounders were tested in the whole model. Non-significant confounders, variables of interest and interactions were removed from the final models. Only significant and trend models are presented in Appendix S5. 

Prior to testing the confounders, we checked for multicollinearity issues among all confounders, as well as among confounders and dependent variables of interest. There were no collinearity issues between any of the factors, thus none of the confounders which we decided to test were removed from the analysis. 

From the factors that differed among our groups, we decided to test Children (categorical, presence or not), Owner Gender (categorical, female or male) and Dog Age (numeric, months of age) as possible confounders, but not FCI Breed Group and Age of Separation from the Mother.

There is quite some literature suggesting that owner/handler gender and dog age can impact dog stress levels (e.g., Buttner et al, 2015; Henessy et al, 1998; Meyer and Forkman, 2014; Shöberl et al, 2016) and hence these two factors were tested as confounders. As regards Children, to our knowledge, there is no study directly studying or showing a relationship between the presence of children in the household and dog stress/welfare. However, a study by Meyer and Forkman (2014) showed that that children in the family was negatively associated with the owners' perception of the relationship with their dogs. As this can eventually result in a negative effect on dog welfare, we also decided to also include Children in the analysis. The reason not to include breed group is motivated by the incoherent relation between breed groups and behavior/temperament. It is known, as you referred, that breeds differ in behavior. However, when one tries to group breeds (either by kennel clubs’ conventional classification like the one we report in the present study, or through genetic relatedness) behavioural divergences of related dog breeds are found (e.g., Turcsán et al 2011). Mixed breeds make the issue even more complex. In order to include breed as a factor we would need to consider “mixed breed” as a breed group level, but that approach ignores the fact that this is a highly heterogenous group, where different individual dogs are related to completely different groups of pure breeds, and where in most cases we do not know which breeds they were crossed with (estimating dog breeds from appearance is not reliable, Voith et al 2009). Finally, we did consider testing Age of Separation from the Mother as a confounder, as it is documented that early separation can lead to stress and stress-associated behaviors in dogs (e..g, Tiira et al, 2012). However, we had a high amount of missing data for this factor (around 22% of the owners reported not knowing when the dog had been separated from the mother), which made it impossible for us to carry on with this test without the possibility of generating questionable results.

Results indicated that Presence of Children was a confounder affecting body turn and body shake in opposite directions. Dog Age was negatively associated with body turn and yawn. Owner Gender did not affect any of the studied variables. Although the significant confounders were kept in their respective models (as they should), we cannot draw any conclusions from an effect of a confounder. It is important to keep in mind that confounders correlate not only with the dependents variables but also with the INDEPENDENT VARIABLES of interest (once their parameter estimates were changed by more than 10% when confounders were added to the models). Thus, no conclusions can be drawn about why and how each of the significant confounders affected the dependent variables, as they are mere confounders, and also some of their results were not consistent among the behaviors. Similarly, we cannot conclude that Owner Gender, for example, is not relevant for explaining the behaviors studied. Our hypothesis did not include these possible confounders as variables of interest and the study was not designed to detect an effect of these, thus the possible confounders can and should only be used to improve the models mathematically. 

To summarize, we appreciate this constructive criticism and wish to make it clear that the inclusion of the confounders improved some of the models explaining the dependent variables as a function of our treatments, while having only minor effects on the results. The effects of our variables of interest (Group and Training Session) remained robust and consistent, as you will find in the manuscript. There were minor adjustments on the p- and Z values. Following the comments by Reviewer 3, we re-analyzed the cognitive bias data with a modeling approach, and were able to test the same confounders to explain latency to reach the bowls. None of the confounders tested were significant in the cognitive bias task. Last, but not least, we added dog ID as a repeated measure in our behavioral (and cognitive bias) analysis, since we had each dog being subjected to 3 training sessions (and to 5 bowl locations). This probably helped to extract some of the variability caused by the unbalanced confounders and improved our models.

2) I think that your statement from your reply regarding ‘de-briefing’ the instructors/owners at the training schools regarding the true goal of the study should be included in your section on Recruitment (2.2.1). 

This information has been included and this section now reads: 

“Dog training schools within the metropolitan area of Porto, Portugal were searched on the internet. Eight schools were selected based on both their geographical proximity and on the listed training methods. Posteriorly, by telephone, head trainers were asked about their willingness to participate in a study to evaluate dog stress and welfare in the context of training and the methodological approach was thoroughly explained. Dog trainers, however, were not made aware that study results were going to be further compared among different training methods, to avoid any biases during training sessions. Of the eight contacted schools, seven agreed to participate. AFTER STUDY CONCLUSION, A DEBRIEFING WITH THE PARTICIPATING TRAINING SCHOOLS WAS PERFORMED IN ORDER TO COMMUNICATE THE RESULTS.”

3) I believe you have adequately dealt with the part of comment #1 of Reviewer #3 regarding baseline cortisol measures (although, as the reviewer correctly points out, there are no ‘baseline’ measures prior to each training session- and this should be acknowledged in the paper).

As we explain in point 1 in the response to Reviewer 3 comments below, there is a baseline physiological measure of stress outside training, through the saliva samples obtained by owners at home on non-training days. This is now mentioned in the last sentence in the 2nd last paragraph of the Discussion: “Whereas we do not know the animals’ stress levels before the start of training, cortisol data shows no differences between training groups on non-training days.”

Reviewer #1 comments

1) The authors did a nice job editing the text for language & overhauling the statistics section. I'm puzzled why a mixed model negative binomial regression was not used to control for the random effects of individual? This is a repeated measures design so partitioning that error is appropriate.

This is correct, thank you for pointing that out. To make sure the repeated measures were necessary in the model, we compared the original models with new final ones with repeated measures (Clarke, 2007). Comparison results indicated that repeated measures were indeed appropriate and improved the original models. Therefore, we modeled the frequency of stress-related behaviors, behavioral states, and panting again as functions of the same treatments (Group and Training Session), their interaction and all possible confounders (as described in the methodology), WITH repeated measures. As you will notice, our new results are similar to the ones we had before and the new parameter estimates were updated in Appendix S5. Only significant and trend models are detailed in Appendix S5.

2) Though they have made many changes to the manuscript, the cognitive bias issue has not been addressed in a substantive way. Because so much is made of assessing long-term welfare, I think it demands a more detailed examination of why they chose to substitute an entirely novel welfare indicator for Phase II without including any of the behavioral or physiological indicators used in Phase I. It bears discussing how a possibly transient negative affective state indicates long-term poor welfare.

We rethought our claims on having assessed long-term welfare with the cognitive bias task, and the current version of the manuscprit advances a more less complex claim. Throughout the manuscript, we now refer to welfare within the training context (where we previously referred to short-term welfare effects) and outside the training context (where we previously referred to long-term welfare effects). Cognitive bias is now treated as an indicator of welfare outside the training context.

3) This is a new comment, but I suggest including means + SEMs in the text of the results section. The graphs are difficult to see, and it seems odd to have to go the supplemental materials to view results used as dependent variables. A summary table with average occurrence by Group would be helpful. 

We have included means and SEMs for the stress-related behaviors as Figure 2 may actually be difficult to read, and also for the cortisol data that is not depicted in Figure 5 (namely, baseline and post-training levels). We considered including a table summarizing all the results, but this would result in redundancy of information. With the figures and the means and SEMs in the text for the data that may be difficult to read in the figures, we think we have adequatly dealt with this concern.

4) It would be nice if the Discussion considered the uniformly low occurrence of stress indicators per training session... while statistically significant one wonders how biologically relevant some of these differences are (e.g. 3 versus 4 "move away")? Similarly I think the authors have captured nice data worthy of further consideration - e.g. what do they tell us about potentially robust indicators of acute stress?

We agree that this is very interesting but in our opinion this study does not allow us to speculate about indicators of acute stress. The levels of stress-related behaviours are comparable to those found by Cooper et al (2014) in a similar study. However, the results of these two studies refer to a very specific context – that of dogs being trained for basic obedience. To establish robust indicators of acute stress would require a different kind of study, where dogs were subject to known stressors of a variety of types. 

Reviewer #3 comments

Major concerns:

1) Authors found that the stress-related behaviours changed across the three groups during the training. But we do not know what the dogs’ level of stress was before training. It is possible, therefore, that other factors may have led to stress-related behaviours. Consider for instance the living conditions of the dogs. Owners that choose specific training centres may have different approaches to the dogs when interacting with them in their home context. Moreover, several factors in training methods influence dogs’ performance and possibly their level of stress, like a tight schedule of reinforcement/punishment, the characteristics of the to-be-punished behaviour, and several other features. There is plenty of literature on the topic (I just mention one review chapter, but there are several others: Hineline, P. N., & Rosales-Ruiz, J. (2013). Behavior in relation to aversive events: Punishment and negative reinforcement. In G. J. Madden, W. V. Dube, T. D. Hackenberg, G. P. Hanley, & K. A. Lattal (Eds.), APA handbooks in psychology®. APA handbook of behavior analysis, Vol. 1. Methods and principles (p. 483–512)) which seems not only to be missing in the Introduction but also in the Authors’ hypothesis, as revealed by the absence of a baseline measurement for each participating dog.

The study design with recruitment through training schools did not allow us to measure how the dogs behaved before they started attending training schools. We agree that other factors will affect stress levels, and we have approached this in different ways to minimize the risk of bias. Most importantly, we obtained baseline data on cortisol levels by instructing owners to take a cortisol sample at home at around the same time of day as training took place but on days where the dogs did not go to the training school (section 2.4.1). Baseline cortisol levels were similar between the three groups, indicating that the baseline levels of stress was not different between dogs being trained with different methods. Furthermore, we collected information about the dog and their living conditions through a questionnaire to the owners (section 2.5) so that we could see how these were distributed over treatment groups. Where a given situation was overrepresented in one treatment group, such as for example owner gender or the presence/absence of children, this was accounted for by including potential confounders in the statistical analysis (see the response to Editor comment 1 above). These considerations are now mentioned in the 2nd last paragraph of the Discussion. Whereas we agree that it would be interesting to consider the effect of ‘tightness’ of the schedule of reinforcement/punishment and stress-related behaviors in dog training, this was beyond the scope of the present study. By focusing on basic obedience training, we ensured that the behaviors under training were approximately the same across schools. 

2) The non-parametric test used for the Judgment Bias Paradigm does not allow to test the Group x Bowl Location interaction. This interaction would attest that the latency to approach the different locations of the bowl changed according to the training conditions. Therefore, it provides stronger evidence in support of your hypothesis. This interaction should be explored. On the topic, see Gygax (2014, doi: 10.1016/j.anbehav.2014.06.013).

Thank you for bringing this up and for the reading suggestion. We used a generalized mixed model (GLMM) to evaluate raw latencies to reach the bowl as a function of our variables of interest (Group and Bowl Location), their interaction, and possible confounders (as described in the methodology), while accounting for the non-constant variability of our data. Raw latencies were used as suggested by Gygagx et al. (2014) and the individual variability among dogs was extracted from the model by adding Dog ID as a random effect, thus there was no need for adjusting the latencies with this approach. As you can see in the Results section, latency to reach the bowl was affected by both Group and Bowl Location, but not by their interaction, which was then removed from the final model. None of the tested confounders (Children, Owner Gender, and Dog Age) had a significant effect, possibly due to the effective blocking of dogs as a random effect, thus confounders were also removed from the final model. Model parameter estimates are presented in Appendix S5 submitted with the present manuscript.

3) It is not clear to me whether the dogs had finished the training before doing the Judgment Bias task or they had just completed phase 1 of your experiment. If they had not completed the training, then it is misleading to claim that the Judgment Bias task assessed long-term welfare. In general, I think that discussing the results in terms of welfare within and outside the training context would be more appropriate because “short-“ and “long-term” are ambiguous concepts.

The dogs had not necessarily finished the training (in fact most of them did not) and this is now made clearer in the text – now it reads “After finishing data collection for Phase 1, dogs participated in Phase 2, which consisted of a spatial cognitive bias task. THE END OF PHASE 1 DID NOT CORRESPOND TO THE CONCLUSION OF THE TRAINING PROGRAMS FOR THE DOGS, AS THIS WOULD RESULT IN DIFFERENT DOGS BEING EXPOSED TO SUBSTANTIALLY DIFFERENT AMOUNTS OF TRAINING BEFORE BEING ASSESSED FOR COGNITIVE BIAS. Instead, for standardization purposes, we ensured that 1) dogs had attended the training school for at least one month prior to Phase 2 and that 2) the cognitive bias task was conducted within one month of completing Phase 1.” 

We have changed the terminology to describe the different aspects of welfare that were measured with the two methods. Throughout the manuscript, we now refer to welfare within the training context (where we previously referred to short-term welfare effects) and outside the training context (where we previously referred to long-term welfare effects). 

Minor points:

4) Could you please report the effect sizes of your results to facilitate future meta-analysis on this topic?

We added a table on Supporting Information reporting the effect sizes (Appendix S4).

Finally, all the minor gramatical suggestions have been accepted.

References

1. Buttner AP, Thompson B, Strasser R, Santo J. Evidence for a synchronization of hormonal states between humans and dogs during competition. Physiol Behav. 2015; 147, 54‐62. 

2. Hennessy MB, Williams MT, Miller D, Douglas CW, Voith VL. Influence of male and female petters on plasma cortisol and behaviour: can human interaction reduce the stress of dogs in a public animal shelter? Appl Anim Behav Sci. 1998; 61(1), 63-77.

3. Meyer I, Forkman B. Dog and owner characteristics affecting the dog–owner relationship. J Vet Behav. 2014; 9(4), 143-150

4. Schöberl I, Wedl M, Beetz A, Kotrschal K. Psychobiological Factors Affecting Cortisol Variability in Human-Dog Dyads. PLoS One. 2017; 12(2), e0170707.

5. Turcsán B, Kubinyi E, Miklósi A. Trainability and boldness traits differ between dog breed clusters based on conventional breed categories and genetic relatedness. Appl Anim Behav Sci. 2011; 132, 61–70.

6. Voith VL, Ingram E, Mitsouras K, Irizarry K. Comparison of adoption agency breed identification and DNA breed identification of dogs. J Appl Anim Welf Sci. 2009; 12(3), 253‐262. 

7. Tiira K, Lohi H. Reliability and validity of a questionnaire survey in canine anxiety research. Appl. Anim. Behav. Sci. 2014; 155, 82–92.

8. Clarke, KA. A simple distribution-free test for nonnested model selection. Political Anal. 2007; 15(3), 347-363.

---

## [Decision Letter · Decision Letter 2]

30 Jun 2020

PONE-D-19-29749R2

Does training method matter? Evidence for the negative impact of aversive-based methods on companion dog welfare

PLOS ONE

Dear Dr. Vieira de Castro,

Thank you for submitting your manuscript to PLOS ONE. After careful consideration, we feel that it has merit but does not fully meet PLOS ONE’s publication criteria as it currently stands. Therefore, we invite you to submit a revised version of the manuscript that addresses the points raised during the review process.

Dear Ana-

Thank-you for all your efforts in revising the paper to date- it has become a much stronger manuscript! I believe that you have made significant improvements in the statistical analyses, in accordance with the advice you received. You defended your decisions regarding the new analyses quite well in your response to the reviewers; however, your approach is not explained quite as well in the actual paper itself. In addition, both I and the two reviewers (one of which, Reviewer 4, is new), have some suggestions and requirements for clarifying the interpretation of your results in the manuscript, as outlined below. Finally, as indicated by Reviewer 4, it is not clear in the manuscript whether there is evidence of strong inter-rater reliability for the behavioural coding you performed. As described below, this section needs to be addressed more fully. These are the reasons for requiring an additional revision.

My hope is that you find the following constructive criticisms useful. I know (from experience!) that once a paper has been revised twice, it can become frustrating for authors to deal with additional changes that are required prior to acceptance. I believe that there are strengths to your study that make a real (and “real world”) contribution to the literature on dog training and welfare. Given the topic, it is likely that your manuscript will be widely read and cited. Therefore, I feel it is of utmost importance that your story as demonstrated by your data be “iron-clad” and solid, and free of any criticism around how the data are analyzed and interpreted, given your study design and methodology. 

**I. Confounds:**

I appreciate your comments in the response to reviewers letter regarding the fact that the confounders that appeared in your study- i.e., that the groups differed in variables other than those you are testing (training method), such as owner gender, dog breed, etc.- were not originally part of your hypotheses, and that you should not test specifically for them. I agree. It is an unfortunate fact that one difficulty in such “real world” research, in which we cannot randomly assign subjects to groups, is the presence of confounders which make our interpretation of our variables of interest tricky. If confounds are present (and they almost always are!), they have to be fully explicated, addressed statistically to the greatest extent possible, and then not forgotten when the data are interpreted. You have done each of these steps to a good degree, but I think a few more additions could improve the manuscript. 

1) First, please consider whether in the Introduction you can introduce the notion that this is (what I call) a “real world” study, and you are not randomly assigning dogs/owners to training schools. Indicate that because of this, you will be evaluating for the presence of specific differences among the training groups that might influence to overall variables of interest. I believe this, in fact, is a strength of your paper, as you have taken considerable efforts to collect and analyze the demographic data for owners and dogs! However, while you mention the “Questionnaire” in the methods section and the data appear in the results, your efforts are not mentioned anywhere in the Introduction. It is fine to not have hypotheses about the questionnaire outcomes, of course, but I believe it would be good to highlight the care that you are taking in collecting such data, and your awareness that such factors could impact your group differences on the variables of interest. This point was also raised in comment #1 by Reviewer #3.

2) It would be useful to shorten what you write in response to the reviewers (under Editor Comments, 1), and put this in Section 2.7, to provide a fuller description of the choices you made regarding how you analyzed the “other” group differences (confounders) in relation to your variables of interest. This might also address the first part of comment #1 by Reviewer #3... i.e., if the training method still affects body turn, shake, yawn, and low state after controlling for potential significant confounders. (However, please see the entire comment and respond appropriately). It also might address the comment on “Statistical analysis” by Reviewer #4. 

3) With respect to this reviewer’s comment on not including breed as a confounder, even though it differed significantly among groups, please address this more fully in the paper if you decide not to analyze it further. Currently, there is only one line in the Discussion (line 711-712) that dismisses the potential breed effect, which I- and likely many other readers- feel might be having an effect.

4) For the results of the “Questionnaire”, I agree with Reviewer #3 that the interpretation some of the differences in dog and owner demographics is difficult without looking in Appendix S3. However, instead of placing the appendix in the main text, I would recommend that for Sections 3.1.1 and 3.1.2, the direction for the significant results be placed in the main text by stating the medians/range (means/sd)– or proportions, whatever is most appropriate for the variable (vs. just the statistic and associated probability for each finding).  

5) Although you describe the possible effects of confounds as limitations in the Discussion, I think more exploration of these variables as alternate possible explanations for some of the findings- OR why they are NOT as strong explanations- is warranted. It is clear that your hypotheses are focussed on training method comparisons, and that should be the main focus of the Discussion. But engaging in some more discussion of how the ‘other’ group differences which appeared might also impact the behaviours during training, the cognitive bias findings, and the cortisol outcome (no difference) could be worthwhile and generate further research ideas. Also, the comment by Reviewer #4 that owner-dog interactions during a training session might reflect daily non-training interactions should be integrated into the Discussion as well. 

**II. Cognitive Bias Outcomes**

The new analysis in line with Gygax’s (2014) recommendations is sound. However, it is not clear to me that your interpretation of its meaning is! As I understand the concept of cognitive bias, it is specifically the difference in behaviour towards the “ambiguous” stimulus (M, in this case) which is interpreted as either a more optimistic or a more pessimistic bias. In your data, the Group Aversive dogs responded to ALL the food bowl locations more slowly, and the latency to bowl M did not change for this group relative to their response to the other bowls. This is interesting indeed and might indicate that in the Group Aversive dogs, there is more behavioural inhibition or the like. However, is it accurate to call this pessimism? Can you please either address this issue in the manuscript and support your interpretation with some citations that also interpret latency to perform “all” (vs. just ambiguous) tasks as a pessimistic bias, or update how this cognitive bias finding is interpreted in the manuscript? 

Currently, given the new cognitive bias findings, I believe there is NOT much solid support for any welfare effect outside the context of training, as the cortisol shows no differences in “non-training day” measures. So, supporting your current interpretation of the cognitive bias outcome is critical for your argument to stand.  If you cannot sufficiently bolster the cognitive bias interpretation as above, it might be necessary to pull back from claims about “poorer welfare” for dogs exposed to aversive training classes, and instead focus on your strong effects, which are the group differences emerging from the behaviour coded during training sessions, and the post-training cortisol levels. 

**III. Inter-rater Reliability**

There is still a lack of clarity on how strong inter-rater reliability (IRR) for the behavioural measures actually is, as pointed out by Reviewer #4.  It is critically important for you to be able to convince readers that there is acceptable/high inter-rater reliability for these behaviours, as the behaviour effects are some of your strongest. Currently, as you report it, there were 3 observers coding videos, only 2 of which were blind to the group assignment of the dogs. The first observer, who was NOT blind to condition, was responsible for coding the vast majority of the videos. This, in and of itself, is not necessarily a problem, IF you can demonstrate convincingly that there is high IRR for each behaviour coded among the observers. This requires reporting: 1) the total number of videos watched/coded and the percentage of videos coded by each observer,  and 2) for each behaviour, a value for IRR (whatever statistic best suits your situation)- which can be presented as an appendix. Without this additional information, we are unable to ascertain the extent to which it is possible that unconscious bias in coding by the non-blinded observer might have influenced the outcome. So, please augment this section. If it requires additional coding by observers, this is worthy investment in time and effort. 

**IV. ****Effect sizes: **

Inclusion of effect sizes is excellent, as pointed out by reviewers. However, they are lost in the Appendix. Please include them in the main text, with each result reported. The magnitude of the effect sizes for the behaviours is a strength!

**V. ****Appendix vs. In-text:**

Both reviewers recommend moving some of the information in the Appendices to the main text. It is my preference that the ethograms/behavioural definitions appear in the main text in a table, not in an appendix. However, for the other appendices, I believe it is “author’s choice”.

As usual, please respond to each of the Reviewer comments in your letter. There are additional recommendations from Reviewer #4 which are quite useful. I encourage you to keep going with this manuscript, as it conrtibutes knowledge not currently in the literature!

Best,

Carolyn

We look forward to receiving your revised manuscript.

Kind regards,

Carolyn J Walsh, PhD

Academic Editor

PLOS ONE

Reviewers' comments:

Reviewer's Responses to Questions

**Comments to the Author**

1. If the authors have adequately addressed your comments raised in a previous round of review and you feel that this manuscript is now acceptable for publication, you may indicate that here to bypass the “Comments to the Author” section, enter your conflict of interest statement in the “Confidential to Editor” section, and submit your "Accept" recommendation.

Reviewer #3: (No Response)

Reviewer #4: (No Response)

2. Is the manuscript technically sound, and do the data support the conclusions?

Reviewer #3: (No Response)

Reviewer #4: Yes

3. Has the statistical analysis been performed appropriately and rigorously? 

Reviewer #3: (No Response)

Reviewer #4: Yes

4. Have the authors made all data underlying the findings in their manuscript fully available?

Reviewer #3: (No Response)

Reviewer #4: Yes

5. Is the manuscript presented in an intelligible fashion and written in standard English?

Reviewer #3: (No Response)

Reviewer #4: Yes

6. Review Comments to the Author

Reviewer #3: 1) I appreciate the fact that you have considered several potential confounders. Since you have found that the three groups differ along several demographic factors, you should report which group is different in the main text - not only in the SI. It is otherwise hard to interpret the results from line 472 to 475. Also, and more importantly, you should check if the training method still affects Body turn, Body shake, Yawn and Low State after controlling for the potential significant confounders.

I am still not convinced that the measurement of cortisol levels during - and not before - Phase 1 can be considered as a reliable baseline. In fact, in Phase 2, you demonstrated that the training method had affected dogs' welfare outside of its immediate context. Why should cortisol levels not be affected in the same way?

Also, I still think it is important for a potential reader to understand that several factors involved in training methods might have affected your outcome variables. These factors may also have affected your results. You should at least mention this in the Introduction.

2) Thanks for your reply, I believe that you have properly addressed this point.

3) Thank you for clarifying this point. Because some dogs had finished the training while others had not, I wonder if the three groups differed in the number of training sessions they had attended before Phase 2.

4) Thank you for adding the effect-sizes.

Reviewer #4: I have read the paper and past reviews with interest and commend the authors on their work. It is indeed difficult to disentangle the many factors potentially affecting dogs’ welfare. While previous studies regarding relationships of owners’ training style and dog welfare have been mostly correlational, this manuscript has several strengths, as designation of training methods, as well as the welfare indicators were done based on objective measures and not owner report. The authors used a multimodal approach – behavioural indicators of acute stress, cortisol measures as well as the judgement bias test. I also appreciate that it takes a lot of effort to recruit and test a sample size of 92 dogs.

The revised statistics appear to be well-founded, and the authors appropriately acknowledge the limitations of their study. Clearly there are many influencing factors that can affect a dog’s daily welfare. Nonetheless, it would not be unreasonable to assume that owners’ interactions during the training session are indicative of their interactions during everyday life, and this could potentially explain the differences in the cognitive bias tests. This concern (according to reviewer 3 of the last round of reviews) is actually something I would view as an advantage, with the results likely not only having implications for the time the dog spends in dog school, but potentially the everyday interactions with their owners. Probably this should be discussed, as different reviewers independently brought this up.

Effect sizes (Cohen’s d) reported in the appendix were large, as many were >1. I think it would be worth pointing out in the main text that there were large effect sizes, which is even more informative than the p values.

Abstract:

One of the study’s strengths, in my opinion, is that training method was objectively measured. Since not everybody reads the whole paper, I would recommend to include this information in the abstract such as was stated in the Introduction “By performing an objective assessment of training methods (through the direct observation of training sessions) and by using objective measures of welfare (behavioral and physiological data to assess effects during training, and a cognitive bias task to assess effects outside training)”

Line 29: I don’t think the authors can claim to have investigated the “entire range of aversive-based techniques (beyond shock-collars)”. Rather, it is relevant that the observed intended positive punishments were presumably less aversive than shock collars, and still clear differences between the groups were found. So I would rather frame it such that previous studies used very highly aversive stimuli such as shock collars which may not be relevant to most dogs’ everyday lives, whereas the observed techniques were.

Line 104: “we addressed the question of whether aversive-based methods actually compromise the well-being of companion dogs”

- Perhaps it would be beneficial to state this in a more neutral way such as “assessed the effects of reward-based and aversive-based methods on welfare of companion dogs”.

Although welfare is unlikely to be influenced by time in the training school alone, it is likely to reflect on the everyday interaction of the dogs and owners

Line 125: term “posteriorly” – I believe you mean “Prior to inclusion in the study”, rather than after?

Line 147: include a reference for the statement “In order to be coherent with the standard for classification of operant conditioning procedures as reinforcement or punishment (which is based not on the procedure itself but on its effect on behavior),

Line 155: I feel it is important how the schools were designated as aversive or reward based, so personally I would prefer to have this information in the main manuscript, rather than the appendix

Line 327: As above, I would prefer to know the details of behaviour codings to assess welfare from the paper, rather than the appendix

Line 337: it is not totally clear to me on the basis of how many videos reliability was assessed at the end, and what percentage of videos was coded by each of the coders

Statistical analysis:

Line 397: Why were confounders tested one at a time and not simply included in the full model? (I realise it might possibly be due to power/ sample size if too many variables are included in the model?).

While I wouldn’t insist on it, in my opinion including breed in the model might be worthwhile. The authors commented that they found doing this not useful given that mixed breeds are not a homogenous group. There are, however some potentially relevant systematic difference also between mixed breeds and purebreds:

Turcsán, B., Miklósi, Á., & Kubinyi, E. (2017). Owner perceived differences between mixed-breed and purebred dogs. PloS One, 12(2), e0172720.

Riemer, S. (2019). Not a one-way road – severity, progression and prevention of firework fears in dogs. Plos One, 14(9), e0218150.

Line 426: Effect sizes could be reported in the results, rather than the appendix

Line 538: maybe “require” instead of “take”? (English suggestions)

Line 619: maybe “possibly reflects” instead of “is possibly a reflex of”

Line 658: also one year since the “treatment” is a long time for this to still have an effect

Discussion: perhaps it could be discussed that the cognitive bias test indicates welfare differences between the three groups, but this was not reflected in baseline cortisol measures

Line 714: However, recent studies show that adoption >8 weeks is also associated with a higher incidence of behaviour problems than adoption at 8 weeks

Jokinen, O., Appleby, D., Sandbacka-Saxén, S., Appleby, T., & Valros, A. (2017). Homing age influences the prevalence of aggressive and avoidance-related behaviour in adult dogs. Applied Animal Behaviour Science.

Puurunen, J., Hakanen, E., Salonen, M. K., Mikkola, S., Sulkama, S., Araujo, C., & Lohi, H. (2020). Inadequate socialisation, inactivity, and urban living environment are associated with social fearfulness in pet dogs. Scientific reports, 10(1), 1-10.

Appendix 1:

I would suggest to write “presumably unpleasant”/ “presumably pleasant” stimulus, rather than having “unpleasant” or “pleasant” in parentheses.

I was wondering how often petting the dog was observed compared to feeding? (as being petted might not necessarily be perceived as pleasant in a training context, even if it is meant as a reward by the human)

I would appreciate a full list of all behaviours included in the definitions of “pleasant” and “unpleasant”, and perhaps their frequencies. Perhaps the current Appendix 1 could go into the main text, and the frequencies of different types of pleasant and unpleasant stimuli in the Appendix.

Appendix 2:

I think in the definition for move away it should read “dog takes” not “dog gives”

The visible lines seem to be slightly mixed up for vocalisations.

Paw lift: “for a brief or a more prolonged time” is very unspecific

Fig 3 and 4 differ in that there are lines in Fig 3 at the x-axis and between labels but not 4 in the same position.

General: I found some double empty spaces in the text, which can be found with the search and replace function.

7. PLOS authors have the option to publish the peer review history of their article (what does this mean?). If published, this will include your full peer review and any attached files.

Reviewer #3: No

Reviewer #4: No

---

## [Author Response · Author response to Decision Letter 2]

10 Aug 2020

RESPONSE TO EDITOR AND REVIEWERS

We appreciate all the constructive criticism provided once again by the editor and the reviewers. Our paper has become admittedly stronger with the review process. For the current version, we added information on the analysis of the confounders to the Statistical analyses section and also included a review of previous studies reporting an effect of these variables on dog behaviour and stress in the Discussion section. Moreover, the cognitive bias results are now discussed in more depth, with references to other studies that found similar results (i.e., differences also for training stimuli). Additionally, as requested, further information on how IRR was calculated is reported and some material that was in the appendices in the previous version of the paper was moved to the main text. All the remaining suggestions were also carefully considered. Detailed responses to all the comments are presented in what follows.

Editor:

I. Confounds

I appreciate your comments in the response to reviewers letter regarding the fact that the confounders that appeared in your study- i.e., that the groups differed in variables other than those you are testing (training method), such as owner gender, dog breed, etc.- were not originally part of your hypotheses, and that you should not test specifically for them. I agree. It is an unfortunate fact that one difficulty in such “real world” research, in which we cannot randomly assign subjects to groups, is the presence of confounders which make our interpretation of our variables of interest tricky. If confounds are present (and they almost always are!), they have to be fully explicated, addressed statistically to the greatest extent possible, and then not forgotten when the data are interpreted. You have done each of these steps to a good degree, but I think a few more additions could improve the manuscript. 

1) First, please consider whether in the Introduction you can introduce the notion that this is (what I call) a “real world” study, and you are not randomly assigning dogs/owners to training schools. Indicate that because of this, you will be evaluating for the presence of specific differences among the training groups that might influence to overall variables of interest. I believe this, in fact, is a strength of your paper, as you have taken considerable efforts to collect and analyze the demographic data for owners and dogs! However, while you mention the “Questionnaire” in the methods section and the data appear in the results, your efforts are not mentioned anywhere in the Introduction. It is fine to not have hypotheses about the questionnaire outcomes, of course, but I believe it would be good to highlight the care that you are taking in collecting such data, and your awareness that such factors could impact your group differences on the variables of interest. This point was also raised in comment #1 by Reviewer #3.

To clarify this, we have added the following two sentences to the second last paragraph of the introduction: 

“We used a quasi-experimental approach in which dog-owner dyads were recruited to participate through the training school at which they were enrolled. As treatment could not be randomized, data on potential confounders was collected to be included in the analysis of treatment effects.”

2) It would be useful to shorten what you write in response to the reviewers (under Editor Comments, 1), and put this in Section 2.7, to provide a fuller description of the choices you made regarding how you analyzed the “other” group differences (confounders) in relation to your variables of interest. This might also address the first part of comment #1 by Reviewer #3... i.e., if the training method still affects body turn, shake, yawn, and low state after controlling for potential significant confounders. (However, please see the entire comment and respond appropriately). It also might address the comment on “Statistical analysis” by Reviewer #4. 

We have added the following information to section 2.7, where the statistics analysis is described: 

“To correct for the unbalanced distribution of potential confounders in the dataset, all known confounders for which sufficient data were available were considered in the analysis as follows. First, each confounder was tested, one at a time, in addition to the variables of interest, to verify if there was a significant relation between the confounder and the response variable, and if the confounder substantially changed the model estimate of the independent variables (i.e., group and training session estimates). This way of testing confounders, in which they are tested one at a time, allows to maintain enough statistical power to verify their significance and influence in the models. If more than one confounder was found to be significant, then all the significant confounders were tested in the whole model. Non-significant confounders, variables of interest and interactions were removed from the final models.”

3) With respect to this reviewer’s comment on not including breed as a confounder, even though it differed significantly among groups, please address this more fully in the paper if you decide not to analyze it further. Currently, there is only one line in the Discussion (line 711-712) that dismisses the potential breed effect, which I- and likely many other readers- feel might be having an effect.

We do not dispute that breed may have an effect on behaviour. Breed differences in behaviour are well established, but this applies to single breeds. Because of the heterogeneity of our sample, dogs were not classified according to single breeds but according to FCI breed groups. Moreover, 34% of the dogs were mixed breeds, mainly unknown. We have expanded on this question in the discussion where the relevant section now reads:

“Two of the potential confounders were not included in the analysis because of insufficient reliable data: breed (34% mixed breeds, mainly unknown) and age of separation from the mother (22% unknown). Breed differences in behavior are well established [43] but the classification of breeds into groups has not been found to systematically correlate with behavioral similarities [e.g., 45], and the large percentage of mixed breed dogs where the actual breeds were unknown further constrains a meaningful analysis of this factor in our sample. Literature shows that both early [e.g., 46] and late [e.g., 47, 48] separation from the mother (before and after 8 weeks-old, respectively) can be associated with stress-related behavioral problems in dogs.” 

4) For the results of the “Questionnaire”, I agree with Reviewer #3 that the interpretation of some of the differences in dog and owner demographics is difficult without looking in Appendix S3. However, instead of placing the appendix in the main text, I would recommend that for Sections 3.1.1 and 3.1.2, the direction for the significant results be placed in the main text by stating the medians/range (means/sd)– or proportions, whatever is most appropriate for the variable (vs. just the statistic and associated probability for each finding). 

The table is a frequency table and many of the variables have 5 or more categories, which makes it impossible to report the data as requested. We decided to follows Reviewer #3’ suggestion and condensed Appendix S3 to a table in the main text (now Table 4).

5) Although you describe the possible effects of confounds as limitations in the Discussion, I think more exploration of these variables as alternate possible explanations for some of the findings- OR why they are NOT as strong explanations- is warranted. It is clear that your hypotheses are focussed on training method comparisons, and that should be the main focus of the Discussion. But engaging in some more discussion of how the ‘other’ group differences which appeared might also impact the behaviours during training, the cognitive bias findings, and the cortisol outcome (no difference) could be worthwhile and generate further research ideas. 

The potential confounders have been accounted for in the statistical analysis (except for breed and age of separation from the mother, where we had insufficient reliable data for a meaningful analysis), which means there is an effect of training method that is independent of the confounding effects of other variables. Whereas it would be misleading to discuss an effect of variables that the study was not designed to evaluate (and for which we cannot thus infer any relation to our dependent variables), we have added a review of previous studies reporting an effect of these variables on measures of dog behaviour and stress. The section of the discussion addressing the potential confounders now reads: “The study was not designed to evaluate the effect of these factors and they were therefore treated as potential confounders in the statistical analysis, in order to account for the possibility that they would affect our results. The effects of training method reported in the study are robust to these confounders. We tested for dog age, presence of children in the household and owner gender, factors which have been shown to potentially affect dog stress and welfare [e.g., 41-44]. The presence of children in the family has been found to be negatively associated with the owners' perception of the relationship with their dogs, in what is to our knowledge the only study addressing how this factor affects dog behavior [43]. Most research into the relationship between dog age and stress indicators has been conducted in senior dogs and consistently shows higher baseline cortisol and higher cortisol responses to stressful stimuli in aged dogs [45, 46]; however, our study did not include any senior dog. Schöberl et al (2017) [44] found cortisol to decrease with increasing age of the dog in adult dogs, whereas Henessy et al (1998) [42] found that the juveniles and adults had higher plasma cortisol levels than puppies. Two of the potential confounders were not included in the analysis because of insufficient reliable data: breed (34% mixed breeds, mainly unknown) and age of separation from the mother (22% unknown). Breed differences in behavior are well established [43] but the classification of breeds into groups has not been found to systematically correlate with behavioral similarities [e.g., 47], and the large percentage of mixed breed dogs where the actual breeds were unknown further constrains a meaningful analysis of this factor in our sample. Literature shows that both early [e.g., 48] and late [e.g., 49, 50] separation from the mother (before and after 8 weeks-old, respectively) can be associated with stress-related behavioral problems in dogs.”

Also, the comment by Reviewer #4 that owner-dog interactions during a training session might reflect daily non-training interactions should be integrated into the Discussion as well. 

See response to Reviewer #4 below.

II. Cognitive Bias Outcomes

The new analysis in line with Gygax’s (2014) recommendations is sound. However, it is not clear to me that your interpretation of its meaning is! As I understand the concept of cognitive bias, it is specifically the difference in behaviour towards the “ambiguous” stimulus (M, in this case) which is interpreted as either a more optimistic or a more pessimistic bias. In your data, the Group Aversive dogs responded to ALL the food bowl locations more slowly, and the latency to bowl M did not change for this group relative to their response to the other bowls. This is interesting indeed and might indicate that in the Group Aversive dogs, there is more behavioural inhibition or the like. However, is it accurate to call this pessimism? Can you please either address this issue in the manuscript and support your interpretation with some citations that also interpret latency to perform “all” (vs. just ambiguous) tasks as a pessimistic bias, or update how this cognitive bias finding is interpreted in the manuscript? 

Currently, given the new cognitive bias findings, I believe there is NOT much solid support for any welfare effect outside the context of training, as the cortisol shows no differences in “non-training day” measures. So, supporting your current interpretation of the cognitive bias outcome is critical for your argument to stand. If you cannot sufficiently bolster the cognitive bias interpretation as above, it might be necessary to pull back from claims about “poorer welfare” for dogs exposed to aversive training classes, and instead focus on your strong effects, which are the group differences emerging from the behaviour coded during training sessions, and the post-training cortisol levels. 

Although affect is hypothesised to exert a greater influence on decision-making under ambiguity (test stimuli: NN, M, NP) than under certainty (training stimuli: N, P), some studies in cognitive bias have found differences for both test and training stimuli (e.g., Deakin, 2018; Horváth et al, 2016; Zidar el at, 2018, see Neville et al 2020 for a review). This type of result has also been interpreted as evidence for differences in the valence of the affective states. The fact that differences can emerge for both training and test stimuli has been proposed to result from the fact that choice in the cognitive bias task depends on two different components of the decision-making process: perceived probability and perceived valuation of rewards (and punishments). An individual may be less likely to make a less “risky” or more “pessimistic” response if they consider the reward to be less probable (or punisher more probable) and/or if they consider the reward to be less valuable (or the punisher more aversive) (Iigaya et al, 2016; Neville et al, 2020). In the current study, Group Aversive displayed higher latencies for all the stimuli (training and test). Therefore, the most likely explanation for our findings is that dogs from Group Aversive considered the food reward less probable (as indicated by the higher latencies to the test cues) and also showed a higher valuation of reward loss relative to win (as indicated by the higher latencies to the P and N bowls) (Iigaya et al, 2016). Because similar findings have been interpreted as indicative of more “pessimistic” responses, we are therefore confident that our results reflect a real difference in the affective states of the dogs from Group Aversive vs. the dogs from Groups Reward and Group Mixed. These ideas appear now in the Discussion section:

“When considering welfare outside the training context, we found that, in the cognitive bias task, dogs from Group Aversive displayed higher latencies for all the stimuli than dogs from Group Reward, with no differences being found between Groups Aversive and Mixed nor between Groups Reward and Mixed. Although affect is hypothesized to exert a greater influence on decision-making under ambiguity (i.e., for the test stimuli: NN, M, NP) than under certainty (i.e., for training stimuli: N, P), other studies in cognitive bias have also found differences for both test and training stimuli [e.g., 32-35, see 35 for a review]. This type of result, with differences found for (at least one of) the training stimuli has also been interpreted as evidence for differences in the valence of the affective states. The fact that differences can emerge for both training and test stimuli has been proposed to result from the fact that choice in the cognitive bias task depends on two different components of the decision-making process: perceived probability and perceived valuation of rewards (and punishments). An individual may be less likely to make a less ‘risky’ or more ‘pessimistic’ response if they consider the reward to be less probable (or punisher more probable) and/or if they consider the reward to be less valuable (or the punisher more aversive) [35,36]. In summary, affective states may influence the responses to both the training and the test stimuli in the cognitive bias task, although different components of the decision-making process may be playing a role. Therefore, the most likely explanation for the present findings is that dogs from Group Aversive considered the food reward less probable (as indicated by the higher latencies to the test stimuli) and also showed a higher valuation of reward loss relative to win (as indicated by the higher latencies to the training stimuli) [36]. Overall, these results indicate that dogs from Group Aversive were in a less positive affective state than dogs from Group Reward.”

Regarding the baseline cortisol levels vs. the cognitive bias findings, please refer to our response to Reviewer 3 comment 1b) below. 

III. Inter-rater Reliability

There is still a lack of clarity on how strong inter-rater reliability (IRR) for the behavioural measures actually is, as pointed out by Reviewer #4. It is critically important for you to be able to convince readers that there is acceptable/high inter-rater reliability for these behaviours, as the behaviour effects are some of your strongest. Currently, as you report it, there were 3 observers coding videos, only 2 of which were blind to the group assignment of the dogs. The first observer, who was NOT blind to condition, was responsible for coding the vast majority of the videos. This, in and of itself, is not necessarily a problem, IF you can demonstrate convincingly that there is high IRR for each behaviour coded among the observers. This requires reporting: 1) the total number of videos watched/coded and the percentage of videos coded by each observer, and 2) for each behaviour, a value for IRR (whatever statistic best suits your situation) - which can be presented as an appendix. Without this additional information, we are unable to ascertain the extent to which it is possible that unconscious bias in coding by the non-blinded observer might have influenced the outcome. So, please augment this section. If it requires additional coding by observers, this is worthy investment in time and effort. 

Our method of video analysis and of IRR calculation followed that performed by Cooper et al (2014). As in their study, “Each observer received training to become familiar with the ethogram developed (…) to allow assessment of inter-observer reliability. Inter-observer reliability was tested by allocating four videos to different observers at an early stage of analysis. Consistency in scoring was assessed by calculating the correlation coefficient r (…) Where r>0.8, it was assumed there was good agreement between observers’ scores and they were reliably following the sampling method. Where there was poor agreement (r<0.8), observers received further training to address inconsistencies.” 

In our study, IRR was calculated for the entire ethogram and not for each behavioural category. We changed the text accordingly to make this information clear and also to report all the additional requested data. Now it reads:

“The second and fourth authors were trained to become familiar with the ethograms and inter-observer reliability was assessed for each ethogram by having the corresponding pair of observers watch and code sets of four videos at an early stage of analysis. Cohen’s Kappa coefficient was calculated for each pair of videos using The Observer XT. After analyzing each set of four videos, if there was poor agreement for any video (r<0.80), the observers received further training. Values of r>0.80 were assumed to indicate strong agreement, and once this level was attained for the four videos of the set, the observers began coding videos independently [9]. A total of 265 videos were coded. For the ethogram for continuous sampling, the analysis of 16 videos was needed before a value of r>0.80 was achieved, whereas for the ethogram for scan sampling, r>0.80 was achieved after analysis of 4 videos. Afterwards, for each ethogram, the remaining videos were distributed randomly between observers, while ensuring that each observer coded a similar percentage of videos from each experimental group. The first author coded 76% of the videos with the ethogram for stress-related behaviors and 64% with the ethogram for overall behavioral state and panting.”

IV. Effect sizes: 

Inclusion of effect sizes is excellent, as pointed out by reviewers. However, they are lost in the Appendix. Please include them in the main text, with each result reported. The magnitude of the effect sizes for the behaviours is a strength!

We have incorporated the effect size into the main text – see also our response to Reviewer 3 comment 3 below. 

V. Appendix vs. In-text:

Both reviewers recommend moving some of the information in the Appendices to the main text. It is my preference that the ethograms/behavioural definitions appear in the main text in a table, not in an appendix. However, for the other appendices, I believe it is “author’s choice”.

The following tables were moved from the Appendices to the main text:

Table S1a (Appendix S1) is now Table 1.

Appendix S2 is now Table 2 and Table 3.

Appendix S3 is now condensed in Table 4.

Reviewer #3: 

a) I appreciate the fact that you have considered several potential confounders. Since you have found that the three groups differ along several demographic factors, you should report which group is different in the main text - not only in the SI. It is otherwise hard to interpret the results from line 472 to 475. 

Please check our answer to concern 4) from the Editor. 

Also, and more importantly, you should check if the training method still affects Body turn, Body shake, Yawn and Low State after controlling for the potential significant confounders.

It is not clear to us what the reviewer’s concern is. However, all the results reported in the current and previous versions of this manuscript refer to the effect of training method after controlling for the confounders.

b) I am still not convinced that the measurement of cortisol levels during - and not before - Phase 1 can be considered as a reliable baseline. In fact, in Phase 2, you demonstrated that the training method had affected dogs' welfare outside of its immediate context. Why should cortisol levels not be affected in the same way?

We appreciate this comment and apologize if through the revision process we have contributed to a misunderstanding of the purpose of the baseline cortisol measure. We never intended the cortisol level on non-training days to be used as an indicator of overall stress level. The purpose of determining saliva cortisol in each dog on non-training days was to be able to determine the acute cortisol response to training. We do not intend to make any claim in the manuscript of having a baseline measure of general stress level, but if you (reviewer or editor) think that this is the case, we would appreciate if you would point out where and we will revise accordingly. 

In addition, it is quite consensual that salivary cortisol is not a reliable measure of long-term stress. Importantly, the effects of chronic stress on physiological data and other responses (such as behavior) can be contradictory [e.g., Pawluski J, Jego P, Henry S, Bruchet A, Palme R, Coste C, et al. (2017) Low plasma cortisol and fecal cortisol metabolite measures as indicators of compromised welfare in domestic horses (Equus caballus). PLoS ONE 12(9): e0182257. https://doi.org/10.1371/journal.pone.0182257].

c) Also, I still think it is important for a potential reader to understand that several factors involved in training methods might have affected your outcome variables. These factors may also have affected your results. You should at least mention this in the Introduction.

We have added a reflection on other factors which may influence dogs in the Introduction, where the 2nd paragraph now reads as follows, referring to Hineline & Rosales-Ruiz (2013) as previously suggested by this reviewer:

“Dog training most often involves the use of operant conditioning principles, and dog training methods can be classified according to the principles they implement: aversive-based methods use mainly positive punishment and negative reinforcement and reward-based methods rely on positive reinforcement and negative punishment [3]. WITHIN A GIVEN TRAINING METHOD, SEVERAL FACTORS MAY INFLUENCE HOW DOGS REACT, SUCH AS THE CHARACTERISTICS OF THE BEHAVIOR UNDER TRAINING AND THE TIMING OF REINFORCEMENT/PUNISHMENT [4]. However, the use of aversive-based training methods per se is surrounded by a heated debate, as studies have linked them to compromised dog welfare [5-10].”

2) Thanks for your reply, I believe that you have properly addressed this point.

3) Thank you for clarifying this point. Because some dogs had finished the training while others had not, I wonder if the three groups differed in the number of training sessions they had attended before Phase 2.

This information is in section 3.3, where it reads:

“Prior to the cognitive bias task, dogs from Group Aversive, Mixed, and Reward attended (M±SEM) 6.29±0.47, 7.14±0.65 and 6.07±0.36 training classes, respectively, with no significant differences observed among groups [H(2)=2.7, p=0.258].”

4) Thank you for adding the effect-sizes.

Reviewer #4: 

1) I have read the paper and past reviews with interest and commend the authors on their work. It is indeed difficult to disentangle the many factors potentially affecting dogs’ welfare. While previous studies regarding relationships of owners’ training style and dog welfare have been mostly correlational, this manuscript has several strengths, as designation of training methods, as well as the welfare indicators were done based on objective measures and not owner report. The authors used a multimodal approach – behavioural indicators of acute stress, cortisol measures as well as the judgement bias test. I also appreciate that it takes a lot of effort to recruit and test a sample size of 92 dogs.

We appreciate this recognition of the strengths of the study – thank you!

2) The revised statistics appear to be well-founded, and the authors appropriately acknowledge the limitations of their study. Clearly there are many influencing factors that can affect a dog’s daily welfare. Nonetheless, it would not be unreasonable to assume that owners’ interactions during the training session are indicative of their interactions during everyday life, and this could potentially explain the differences in the cognitive bias tests. This concern (according to reviewer 3 of the last round of reviews) is actually something I would view as an advantage, with the results likely not only having implications for the time the dog spends in dog school, but potentially the everyday interactions with their owners. Probably this should be discussed, as different reviewers independently brought this up.

We appreciate this comment and agree that one would expect that the way owners interact with their dogs during training sessions reflect at least to some extent how they interact during everyday life. Indeed, it is usually part of the training schools’ objective that owners continue to implement what they learn during lessons in all kinds of interactions with their dog. As this reviewer observes in the previous paragraph, the present study is centred on objective measures collected by the research team, and we did not collect data on dog-owner interactions outside the training school. Therefore, we would rather not speculate on the relation between dog-human interactions inside and outside the school as regards our sample. We recognize that the issue is important and have added a reflection in the last paragraph of the discussion, which now ends “This applies not only to training in a formal school setting but whenever owners use reinforcement or punishment in their interactions with the dog.”

3) Effect sizes (Cohen’s d) reported in the appendix were large, as many were >1. I think it would be worth pointing out in the main text that there were large effect sizes, which is even more informative than the p values.

We have incorporated effect size as Cohen’s d in the results section, so that each comparison now reads something like “Group A was higher/lower than Group B (Z=4.6, p<0.001, d=1.02).

Abstract:

One of the study’s strengths, in my opinion, is that training method was objectively measured. Since not everybody reads the whole paper, I would recommend to include this information in the abstract such as was stated in the Introduction “By performing an objective assessment of training methods (through the direct observation of training sessions) and by using objective measures of welfare (behavioral and physiological data to assess effects during training, and a cognitive bias task to assess effects outside training)”

We appreciate this comment, but the word limit for the abstract does not allow more information to be added. The data collection method is already described in detail in the abstract.

Line 29: I don’t think the authors can claim to have investigated the “entire range of aversive-based techniques (beyond shock-collars)”. Rather, it is relevant that the observed intended positive punishments were presumably less aversive than shock collars, and still clear differences between the groups were found. So I would rather frame it such that previous studies used very highly aversive stimuli such as shock collars which may not be relevant to most dogs’ everyday lives, whereas the observed techniques were.

This part of the text has been removed as the abstract was condensed. 

Line 104: “we addressed the question of whether aversive-based methods actually compromise the well-being of companion dogs” - Perhaps it would be beneficial to state this in a more neutral way such as “assessed the effects of reward-based and aversive-based methods on welfare of companion dogs”. Although welfare is unlikely to be influenced by time in the training school alone, it is likely to reflect on the everyday interaction of the dogs and owners

This has been reworded to “assessed the effects of reward-based and aversive-based methods on companion dog welfare”

Line 125: term “posteriorly” – I believe you mean “Prior to inclusion in the study”, rather than after?

This has been reworded so that the description of recruitment now reads “The head trainers were invited by telephone to participate in the study. They were informed that the aim was to evaluate dog stress and welfare in the context of training and the methodological approach was thoroughly explained. To avoid bias during recorded training sessions, the trainers were not made aware that study results were going to be further compared among different training methods.”

Line 147: include a reference for the statement “In order to be coherent with the standard for classification of operant conditioning procedures as reinforcement or punishment (which is based not on the procedure itself but on its effect on behavior).

We have included a reference to Skinner (1953) – Science and Human Behavior.

Line 155: I feel it is important how the schools were designated as aversive or reward based, so personally I would prefer to have this information in the main manuscript, rather than the appendix.

In order to include more detail on how schools were designated, we have incorporated what was previously Table S1 into the main text, as Table 1. 

Line 327: As above, I would prefer to know the details of behaviour codings to assess welfare from the paper, rather than the appendix.

In order to provide details of how behaviour was coding, we have incorporated what was previously Tables S2a and S2b into the main text, as Tables 2 and 3.

Line 337: it is not totally clear to me on the basis of how many videos reliability was assessed at the end, and what percentage of videos was coded by each of the coders.

Please check the answer to the Editor’s comment above.

Statistical analysis

Line 397: Why were confounders tested one at a time and not simply included in the full model? (I realise it might possibly be due to power/ sample size if too many variables are included in the model?).

The reviewer is correct. Now this is directly explained in the text in section 2.7, where it reads:

“This way of testing confounders, in which they are tested one at a time, allows to maintain enough statistical power to verify their significance and influence in the models. If more than one confounder was found to be significant, then all the significant confounders were tested in the whole model. Non-significant confounders, variables of interest and interactions were removed from the final models.”

While I wouldn’t insist on it, in my opinion including breed in the model might be worthwhile. The authors commented that they found doing this not useful given that mixed breeds are not a homogenous group. There are, however some potentially relevant systematic difference also between mixed breeds and purebreds: Turcsán, B., Miklósi, Á., & Kubinyi, E. (2017). Owner perceived differences between mixed-breed and purebred dogs. PloS One, 12(2), e0172720.

Riemer, S. (2019). Not a one-way road – severity, progression and prevention of firework fears in dogs. Plos One, 14(9), e0218150.

We appreciate this comment and do not dispute that breed may affect behaviour. However, a meaningful analysis of a breed effect on behaviour requires a categorization of dogs into breeds for which there is consistent information on behaviour. Our sample is too heterogeneous for a classification into single breeds (many of which are represented by only one dog), and so we used the FCI breed categories. Classification of breeds into groups has not been found to systematically correlate with behavioral similarities, and the large percentage of mixed breed dogs where the actual breeds were unknown further constrains a meaningful analysis of this factor in our sample, as we are explaining in the discussion.

Line 426: Effect sizes could be reported in the results, rather than the appendix

This has been changed – see also our response to Reviewer 3 comment 3 above. 

Line 538: maybe “require” instead of “take”? (English suggestions)

This has been changed as suggested.

Line 619: maybe “possibly reflects” instead of “is possibly a reflex of”

This has been changed as suggested.

Line 658: also one year since the “treatment” is a long time for this to still have an effect

It is not clear to us if the reviewer is suggesting a change to the text here.

Discussion: perhaps it could be discussed that the cognitive bias test indicates welfare differences between the three groups, but this was not reflected in baseline cortisol measures

As explained in detail in our response to Reviewer 3 comment 1b) above, baseline cortisol data in this study was collected to estimate the acute cortisol response to training and not to estimate general stress level. Additionally, check the cited reference for salivary cortisol not being considered as a reliable measure of long-term stress.

Line 714: However, recent studies show that adoption >8 weeks is also associated with a higher incidence of behaviour problems than adoption at 8 weeks

Jokinen, O., Appleby, D., Sandbacka-Saxén, S., Appleby, T., & Valros, A. (2017). Homing age influences the prevalence of aggressive and avoidance-related behaviour in adult dogs. Applied Animal Behaviour Science.

Puurunen, J., Hakanen, E., Salonen, M. K., Mikkola, S., Sulkama, S., Araujo, C., & Lohi, H. (2020). Inadequate socialisation, inactivity, and urban living environment are associated with social fearfulness in pet dogs. Scientific reports, 10(1), 1-10.

Thank you for these useful references. We have reworded this sentence to “Literature shows that both early [e.g., 48] and late [e.g., 49, 50] separation from the mother (before and after 8 weeks-old, respectively) can be associated with stress-related behavioral problems in dogs.”

Appendix 1:

I would suggest to write “presumably unpleasant”/ “presumably pleasant” stimulus, rather than having “unpleasant” or “pleasant” in parentheses.

This has been changed in what is now Table 1 in the manuscript.

I was wondering how often petting the dog was observed compared to feeding? (as being petted might not necessarily be perceived as pleasant in a training context, even if it is meant as reward by the human)

This is a really interesting comment and we totally agree. Although we haven’t formally analysed this data, School A exclusively used petting as R+ as opposed to the remaining schools for which food was the R+ of election (petting was only occasional). This most likely explains why this school had the highest levels of stress behaviors among schools using aversive-based methods (data not presented do). We now address this issue in the Discussion. See next comment.

I would appreciate a full list of all behaviours included in the definitions of “pleasant” and “unpleasant”, and perhaps their frequencies. Perhaps the current Appendix 1 could go into the main text, and the frequencies of different types of pleasant and unpleasant stimuli in the Appendix.

Part of Appendix 1 is already placed as a table in the main text. We have also added to the table the full list of stimuli used in training (we believe this was what the reviewer meant, not behaviours). Additionally, we added the following in the Discussion:

“Moreover, our results suggest that the proportion of aversive stimuli used in training plays a greater role on dogs’ stress levels than the specific training tools used. As an example, one school from Group Mixed used pinch and e-collars, whereas other school from Group Aversive only used choke collars during training. Although the tools used by the former school may be perceived as more aversive, the frequency of stress behaviors was higher in dogs being trained at the latter school. The type of (intended) positive reinforcers also appears to be relevant. All schools except the aforementioned school from Group Aversive used primarily food treats as rewards, whereas the latter only used petting. Although this was not the school using the highest proportion of aversive stimuli, it was the school whose dogs showed the highest frequency of stress behaviors (data not shown). Previous research has shown that petting is a less effective reward than food in training [40]. Having a highly valuable reward might thus be important in reducing stress when aversive stimuli are used in training. The goal of the present study was to test the overall effect on dog welfare of aversive- and reward-based methods as they are used in the real world, but it may be interesting for future studies to focus on disentangling the effects of the different types of stimuli used in training (as has been done with e-collars) [e.g., 9, 25].”

In order to prevent the individual schools from being identifiable through their specific training techniques, we decided not to mention to which specific schools we are referring to in this paragraph.

Appendix 2:

I think in the definition for move away it should read “dog takes” not “dog gives”

This has been changed in the table, which is now Table 2 in the main text. 

The visible lines seem to be slightly mixed up for vocalisations.

We are sorry but we cannot figure out what the reviewer is referring to. We would be thankful if you could point it out so that we can make the corrections accordingly.

Paw lift: “for a brief or a more prolonged time” is very unspecific.

This was changed to “One fore limb only is lifted, usually in a slow movement, and either immediately returned to rest on the ground or remaining lifted for a brief period.”, as the new definition is more specific and still reflects what was agreed between the two observers that coded the video with this ethogram.

Fig 3 and 4 differ in that there are lines in Fig 3 at the x-axis and between labels but not 4 in the same position.

Once again, we are sorry but we cannot figure out what the reviewer is referring to, as in both figures the lines separate the results of the three groups. We would be thankful if you could point it out so that we can make the corrections accordingly

General: I found some double empty spaces in the text, which can be found with the search and replace function.

Thank you for pointing this out; the extra empty spaces have been deleted.

---

## [Decision Letter · Decision Letter 3]

27 Aug 2020

PONE-D-19-29749R3

Does training method matter? Evidence for the negative impact of aversive-based methods on companion dog welfare

PLOS ONE

Dear Dr. Vieira de Castro,

Thank you for submitting your manuscript to PLOS ONE. After careful consideration, we feel that it has merit but does not fully meet PLOS ONE’s publication criteria as it currently stands. Therefore, we invite you to submit a revised version of the manuscript that addresses the points raised during the review process.

Dear Ana-

I completely agree with the reviewer that this paper is important to the growing literature around ethical training and also wish to not create further delays in having it published.

However, I also completely agree with the reviewer that you haven't actually sufficiently addressed the criticisms around inter-rater reliability (IRR). Although the methods you used to evaluate IRR are quite transparent, I believe that they are barely sufficient/likely insufficient to convince skeptics that there has been no bias in the coding of behaviours (as the researcher who coded the majority of behaviours was not blind to training condition of the dogs). This can be fairly easily rectified, with the suggestions made in the last round of revisions and those made currently by Reviewer #4. As an author who has also been asked by reviewers/editors to re-evaluate IRR measures and report them in more detail (i.e., by behaviour), I understand that although this is a straight-forward and relatively easy exercise, it is also frustrating and involves coordinating new work of at least one other coder. However, please re-consider doing so to increase the strength of your IRR reporting in accordance with these suggestions, as it will close a 'window of vulerability' regarding readers' confidence in your outcome- which, in some circles, might be controversial!

As well, please action and/or respond to the other suggestions of Reviewer #4. 

Finally, one of the prior reviewers suggested that I ask you to please take a look at the various affiliations given to the authors on the title page; are these affiliation histories OR are authors' affiliations current (e.g., cross-appointments)? I recommend limiting affiliations to those held when the research was conducted, along with "current" address, if the affiliation has changed. For reference, here is the instruction from the PLoS One website (which I am sure you are familiar):

Each author on the list must have an affiliation. The affiliation includes department, university, or organizational affiliation and its location, including city, state/province (if applicable), and country. Authors have the option to include a current address in addition to the address of their affiliation at the time of the study. The current address should be listed in the byline and clearly labeled “current address.” At a minimum, the address must include the author’s current institution, city, and country.

If an author has multiple affiliations, enter all affiliations on the title page only. In the submission system, enter only the preferred or primary affiliation. Author affiliations will be listed in the typeset PDF article in the same order that authors are listed in the submission.

<o:p></o:p>

<o:p></o:p>

<o:p></o:p>

Looking forward to your response (which I do not anticipate sending back to reviewers),

Carolyn

We look forward to receiving your revised manuscript.

Kind regards,

Carolyn J Walsh, PhD

Academic Editor

PLOS ONE

Reviewers' comments:

Reviewer's Responses to Questions

**Comments to the Author**

1. If the authors have adequately addressed your comments raised in a previous round of review and you feel that this manuscript is now acceptable for publication, you may indicate that here to bypass the “Comments to the Author” section, enter your conflict of interest statement in the “Confidential to Editor” section, and submit your "Accept" recommendation.

Reviewer #4: (No Response)

2. Is the manuscript technically sound, and do the data support the conclusions?

Reviewer #4: Yes

3. Has the statistical analysis been performed appropriately and rigorously? 

Reviewer #4: Yes

4. Have the authors made all data underlying the findings in their manuscript fully available?

Reviewer #4: No

5. Is the manuscript presented in an intelligible fashion and written in standard English?

Reviewer #4: Yes

6. Review Comments to the Author

Reviewer #4: This is a great paper, describing an important piece of work and including a lot of interesting information. Also the discussion has still gained much from the revision and is exciting to read.

I still feel a bit uneasy about the inter rater reliability as it is my opinion that this should generally be done for each coded behavior individually, rather than lumping all together, and as using four videos of over 200 for IRR does not seem much. Nonetheless, I would not wish to delay the publication of this paper any further.

I only have a few more comments

Line 469 3.2.1.1. Stress-related behaviors -> I would prefer the results in a table, making the text easier to read, but this is of course my personal preference.

Line 640: “the use of both shock collars [9] and other negative reinforcement techniques [10]”

-> I know shock collars can be used for negative reinforcement, but aren’t they most often used for positive punishment? So it looked a bit weird to me to say “shock collars and other negative reinforcement techniques” (although I know the terminology is sometimes inconsistent)

Line 732 : This seems to suggest that, although dogs trained in ‘least aversive’ schools

-> this could be a little confusing, as actually the pos reinforcement school was the least aversive school. Maybe just refer to “low aversive” and “highly aversive” schools; or refer to “mixed” as you did in the rest of the text?

Line 788: Presently, there is a lack of scientific evidence regarding the efficacy of different training methods [3], which limits the extent of evidence-based recommendations

->Here this recent study could be cited which supports higher effectiveness of pos. reinforcement methods:

China, L., Mills, D. S., & Cooper, J. J. (2020). Efficacy of Dog Training With and Without Remote Electronic Collars vs. a Focus on Positive Reinforcement. Frontiers in Veterinary Science, 7, 508.

Figure 5: unlike in the other figures, there is no x axis and the names of the different columns are not written in boxes. I.e., the style is different from the other figures.

I may have missed it, but I didn't find where the dataset can be accessed.

7. PLOS authors have the option to publish the peer review history of their article (what does this mean?). If published, this will include your full peer review and any attached files.

Reviewer #4: No

---

## [Author Response · Author response to Decision Letter 3]

7 Oct 2020

RESPONSE TO EDITOR AND REVIEWERS

We are hereby addressing the last round of proposed revisions to our manuscript. Importantly, we believe that the sense of vulnerability surrounding our behavior analysis is explained by the fact that we had not described our process of determining and ensuring inter-rater reliability in sufficient detail so far. In what follows, we present detailed responses to this and all the remaining comments and suggestions. 

Editor:

1. I completely agree with the reviewer that this paper is important to the growing literature around ethical training and also wish to not create further delays in having it published. However, I also completely agree with the reviewer that you haven't actually sufficiently addressed the criticisms around inter-rater reliability (IRR). Although the methods you used to evaluate IRR are quite transparent, I believe that they are barely sufficient/likely insufficient to convince skeptics that there has been no bias in the coding of behaviours (as the researcher who coded the majority of behaviours was not blind to training condition of the dogs). This can be fairly easily rectified, with the suggestions made in the last round of revisions and those made currently by Reviewer #4. As an author who has also been asked by reviewers/editors to re-evaluate IRR measures and report them in more detail (i.e., by behaviour), I understand that although this is a straightforward and relatively easy exercise, it is also frustrating and involves coordinating new work of at least one other coder. However, please re-consider doing so to increase the strength of your IRR reporting in accordance with these suggestions, as it will close a 'window of vulnerability' regarding readers' confidence in your outcome- which, in some circles, might be controversial! 

We appreciate and share the concern that results should be as strong and reliable as possible. Over the course of the exchange of views on this issue, we have realized that we have not described our process of determining inter-rater reliability in sufficient detail, which may be causing this sense of vulnerability. 

The formal calculation of IRR was done using The Observer XT software, which we also used to analyze our videos. Within the software, we chose to use Cohen’s Kappa, which has been recognized as the strongest measure for IRR in animal behavior studies due to the fact that it controls for chance agreement. It happens that The Observer only allows the calculation of IRR for the entire ethogram/list of behaviors. And although we do not dispute that it is relevant to calculate IRR for each behavior individually, the fact is that some authors proposed its calculation for the entire ethogram/list of behaviors (see, for example, Jansen et al, 2003). Given that The Observer is a widely used software for animal behavior studies and that its way of calculating IRR has been backed up by the literature (Jansen et al, 2003), we are confident in the way we addressed IRR. 

We would like to stress that the IRR calculation performed by The Observer does not imply lumping all behaviors together. It involves the calculation of a confusion matrix (see Example 1). 

Additionally, IRR calculation should not be seen as an isolated measure but is part of a wider strategy including training and joint discussions of definitions in order to reach an acceptable level of agreement. To reach an r>0.8 was quite demanding for our continuous sampling ethogram. Training lasted for around 2 months before agreement was reached (i.e., before an r>0.8 could be achieved for a new set of four videos). Importantly, in videos where there were few stress-related behaviors occurring, disagreements on a couple of behaviors would be enough to result in an r<0.8 (see Example 2). This means that a high level of agreement between observers was actually required before starting to code independently. 

Training was lengthy and intensive, and to reach an r>0.8 in a new set of four videos was highly demanding and required a very strong agreement between observers, as opposed to what may appear from the way we reported the information in the previous version of the manuscript. In order to make all the training and IRR assessment process clearer, the manuscript now reads:

 “The second and fourth authors were trained to become familiar with the ethograms and inter-observer reliability was assessed for each ethogram by having the corresponding pair of observers watch and code sets of four videos at an early stage of analysis [9]. Cohen’s Kappa coefficient was calculated for each pair of videos using The Observer XT. After analyzing each set of four videos, if there was poor agreement for any video (r<0.80), the observers checked and discussed all the inconsistencies and, if needed, decided on how to refine the description of the ethogram behaviors. After this, they re-analyzed the same videos and the process was repeated until r>0.8 was achieved for the entire set of videos. Once this level was attained, the observers analyzed a new set of four videos. The whole process was repeated until a value of r>0.8 was achieved for the four videos of a new set in the first attempt (i.e., without the need for a re-analysis). At this point, the observers were assumed to be in strong agreement and began coding videos independently [9]. A total of 265 videos were coded. For the ethogram for continuous sampling, the analysis of 16 videos was needed before agreement was achieved, whereas for the ethogram for scan sampling, agreement was achieved only after the analysis of four videos.”

For the ethogram for scan sampling, an r<0.8 was achieved much quicker: after the first batch of four videos. This shows, in our opinion, that this ethogram was very well described/detailed from the beginning or that it was simply easier to identify the behaviors in question. At the same time, this was not a total surprise, as the same happened in Cooper et al (2014): “Inter-observer reliability was tested by allocating four videos to different observers at an early stage of analysis. (…) Where there was poor agreement (r<0.8), observers received further training to address inconsistencies. This was only necessary for one observer, who following retraining and re-analysis of early tapes was in good agreement with all other observers for the rest of data collection.”

Regarding the number of videos on which we based our IRR calculation, we agree that more than four videos would make the case stronger. When we decided on this approach, we followed the methodology of Cooper et al (2014), a paper that we find of very high quality and published in PLOS ONE a couple of years before the start of our work. We note that Coelho and Bramblett (1981) “found it necessary to train (...) observers between 3 and 4 months in order to achieve a consistent 0.90 level of inter-observer agreement during the training period. However, once this level of agreement was attained, it was maintained for the duration of the 22-month post-training period”. 

We understand that the way we originally presented the IRR determination may have given the impression of a very limited strategy. We hope that the explanation we have now given shows clearly that the IRR calculation across 16 and 4 videos, respectively, is only one part of a larger strategy where the focus is on training and adjustment to obtain agreement. We believe we have demonstrated convincingly that there was high IRR among the observers using the approach for IRR calculation available through one of the leading behavior analysis software packages. The fact that our “behavior” results are in accordance with the cortisol and cognitive bias results of the present study makes, in our opinion, our case even stronger.

This can be fairly easily rectified, with the suggestions made in the last round of revisions and those made currently by Reviewer #4.

These suggestions have already been incorporated in the manuscript, except for the IRR per behavior, for which the justification was presented above.

If you still do not feel our analysis is strong enough, we kindly ask you to clarify your suggestion for recoding with the involvement of a new coder, as it is not clear to us. Recoding data with the people involved in the first coding will not be possible as just one of us (ACVC) is still currently at our research group. This data analysis was finished more than one year ago. We wish to highlight that recoding will always be a lengthy procedure, as this is skilled behavior analysis for which a new coder will need several months of training. 

References:

Jansen RG, Wiertz LF, Meyer ES, Noldus, LPJJ. Reliability analysis of observational data: Problems, solutions, and software implementation. Behavior Research Methods, Instruments & Computers. 2003; 35, 391-399.

Cooper JJ, Cracknell N, Hardiman J, Wright H, Mills D. The welfare consequences and efficacy of training pet dogs with remote electronic training collars in comparison to reward based training. PLoS One. 2014; 9, e102722. 

Coelho AM, Bramblett CA. Interobserver agreement on a molecular ethogram of the genus Papio. Animal Behavior. 1981; 29, 443-448.

2. Finally, one of the prior reviewers suggested that I ask you to please take a look at the various affiliations given to the authors on the title page; are these affiliation histories OR are authors' affiliations current (e.g., cross-appointments)? I recommend limiting affiliations to those held when the research was conducted, along with "current" address, if the affiliation has changed. For reference, here is the instruction from the PLoS One website (which I am sure you are familiar): 

Each author on the list must have an affiliation. The affiliation includes department, university, or organizational affiliation and its location, including city, state/province (if applicable), and country. Authors have the option to include a current address in addition to the address of their affiliation at the time of the study. The current address should be listed in the byline and clearly labeled “current address.” At a minimum, the address must include the author’s current institution, city, and country. 

If an author has multiple affiliations, enter all affiliations on the title page only. In the submission system, enter only the preferred or primary affiliation. Author affiliations will be listed in the typeset PDF article in the same order that authors are listed in the submission. 

The affiliations presented in the title page do indeed refer to those held when the research was conducted. However, this time, only the primary affiliation will be entered in the submission system.

#Reviewer 4:

1. I still feel a bit uneasy about the inter rater reliability as it is my opinion that this should generally be done for each coded behavior individually, rather than lumping all together, and as using four videos of over 200 for IRR does not seem much. Nonetheless, I would not wish to delay the publication of this paper any further. 

As explained in the response to the Editor above, IRR is only one part of a more comprehensive strategy to ensure consistency between observers. The ethogram-based IRR analysis does not consist in lumping all behaviors together, but uses a matrix approach as described above. 

2. Line 469 3.2.1.1. Stress-related behaviors -> I would prefer the results in a table, making the text easier to read, but this is of course my personal preference. 

Please refer to our second letter of response to reviewers:

“Reviewer #1

3) This is a new comment, but I suggest including means + SEMs in the text of the results section. The graphs are difficult to see, and it seems odd to have to go the supplemental materials to view results used as dependent variables. A summary table with average occurrence by Group would be helpful. 

We have included means and SEMs for the stress-related behaviors as Figure 2 may actually be difficult to read, and also for the cortisol data that is not depicted in Figure 5 (namely, baseline and post-training levels). We considered including a table summarizing all the results, but this would result in redundancy of information. With the figures and the means and SEMs in the text for the data that may be difficult to read in the figures, we think we have adequately dealt with this concern.”

3. Line 640: “the use of both shock collars [9] and other negative reinforcement techniques [10]” -> I know shock collars can be used for negative reinforcement, but aren’t they most often used for positive punishment? So it looked a bit weird to me to say “shock collars and other negative reinforcement techniques” (although I know the terminology is sometimes inconsistent) 

We agree and have changed this to “shock collars and other aversive techniques”.

4. Line 732: This seems to suggest that, although dogs trained in ‘least aversive’ schools -> this could be a little confusing, as actually the pos reinforcement school was the least aversive school. Maybe just refer to “low aversive” and “highly aversive” schools; or refer to “mixed” as you did in the rest of the text? 

We agree and have changed accordingly.

5. Line 788: Presently, there is a lack of scientific evidence regarding the efficacy of different training methods [3], which limits the extent of evidence-based recommendations ->Here this recent study could be cited which supports higher effectiveness of pos. reinforcement methods: 

China, L., Mills, D. S., & Cooper, J. J. (2020). Efficacy of Dog Training With and Without Remote Electronic Collars vs. a Focus on Positive Reinforcement. Frontiers in Veterinary Science, 7, 508. 

We added a reference to this recently published paper. This section now reads:

“Presently, the scientific literature on the efficacy of the different methodologies is scarce and inconsistent [3]. Whereas some studies suggest a higher efficacy of reward methods [5, 12, 51-53], one points in the opposite direction [31] and three show no differences between methods [9, 54, 55]. This limits the extent of evidence-based recommendations.”

References:

China, L., Mills, D. S., & Cooper, J. J. (2020). Efficacy of Dog Training With and Without Remote Electronic Collars vs. a Focus on Positive Reinforcement. Frontiers in Veterinary Science, 7, 508. 

Blackwell, E.J., Bolster, C., Richards, G., Loftus, B.A., Casey, R.A. (2012). The use of electronic collars for training domestic dogs: estimated prevalence, reasons and risk factors for use, and owner perceived success as compared to other training methods. BMC Veterinary Research, 8, 93. 

Haverbeke, A., Laporte, B., Depiereux, E., Giffroy, J.M., Diederich, C. (2008). Training methods of military dog handlers and their effects on the team’s performances Applied Animal Behavior Science, 113: 110–122. 

Haverbeke, A., Messaoudi, F., Depiereux, E., Stevens, M., Giffroy, J.-M., Diederich, C., (2010). Efficiency of working dogs undergoing a new Human Familiarization and Training Program. Journal of Veterinary Behavior: Clinical Appications and Research, 5: 112–119. 

Hiby, E.F., Rooney, N.J., Bradshaw, J.W.S. (2004). Dog training methods: their use, effectiveness and interaction with behaviour and welfare. Animal Welfare, 13: 63–69.

Salgirli, Y., Schalke, E., Hackbarth, H. (2012). Comparison of learning effects and stress between 3 different training methods (electronic training collar, pinch collar and quitting signal) in Belgian Malinois Police Dogs. Revue de Médecine Véterinaire, 163, 530–535. 

Sankey, C., Richard-Yris, M.-A., Henry, S., Fureix, C., Nassur, F., Hausberger, M. (2010). Reinforcement as a mediator of the perception of humans by horses (Equus caballus). Animal Cognition, 13: 753–764. 

Visser, E.K., VanDierendonck, M., Ellis, A.D., Rijksen, C, Van Reenen, C.G. (2009). A comparison of sympathetic and conventional training methods on responses to initial horse training. The Veterinary Journal, 181: 48–52. 

6. Figure 5: unlike in the other figures, there is no x axis and the names of the different columns are not written in boxes. I.e., the style is different from the other figures. 

In the current version of the paper, we aimed at a higher figure standardization. Currently, all bar charts have no x or y-axis. We kept the axes visible in Figure 6 (scatterplot) because otherwise the chart would become difficult to read. Boxes naming the different columns were kept only in Figure 5, because this this chart displays data from three training sessions per Group and it becomes easier to read this way. In our opinion, the remaining charts are cleaner and easier to read as they currently stand.

7. I may have missed it, but I didn't find where the dataset can be accessed. 

Thank you for pointing this out - this information was actually accidentally missing in the text. The dataset can be found in the Supporting Information, Appendix S2. Now, the Statistical Analysis section refers: “The entire dataset is available in Appendix S2”.

---

## [Editor Report · Decision Letter 4]

2 Nov 2020

Does training method matter? Evidence for the negative impact of aversive-based methods on companion dog welfare

PONE-D-19-29749R4

Dear Dr. Vieira de Castro,

We’re pleased to inform you that your manuscript has been judged scientifically suitable for publication and will be formally accepted for publication once it meets all outstanding technical requirements.

Kind regards,

Carolyn J Walsh, PhD

Academic Editor

PLOS ONE

Additional Editor Comments (optional):

Thanks for persevering through the revision process!

Best wishes,

Carolyn
---

## [Editor Report · Acceptance letter]

24 Nov 2020

PONE-D-19-29749R4 

Does training method matter? Evidence for the negative impact of aversive-based methods on companion dog welfare 

Dear Dr. Vieira de Castro:

I'm pleased to inform you that your manuscript has been deemed suitable for publication in PLOS ONE. Congratulations! Your manuscript is now with our production department. 

Kind regards, 

on behalf of

Dr. Carolyn J Walsh 

Academic Editor

PLOS ONE